# Cryo-electron tomography reveals how COPII assembles on cargo-containing membranes

Euan Pyle [1,2,3,5], Elizabeth A. Miller[4,6] & Giulia Zanetti [1,2,3] ✉

Proteins traverse the eukaryotic secretory pathway through membrane trafficking between organelles. The coat protein complex II (COPII) mediates the anterograde transport of newly synthesized proteins from the endoplasmic reticulum, engaging cargoes with a wide range of size and biophysical properties. The native architecture of the COPII coat and how cargo might influence COPII carrier morphology remain poorly understood. Here we reconstituted COPII-coated membrane carriers using purified *Saccharomyces cerevisiae* proteins and cell-derived microsomes as a native membrane source. Using cryo-electron tomography with subtomogram averaging, we demonstrate that the COPII coat binds cargo and forms largely spherical vesicles from native membranes. We reveal the architecture of the inner and outer coat layers and shed light on how spherical carriers are formed. Our results provide insights into the architecture and regulation of the COPII coat and advance our current understanding of how membrane curvature is generated.

Eukaryotic cells use the secretory pathway to transport proteins and lipids to their required locations within and outside the cell. Approximately one in three proteins is translocated in the endoplasmic reticulum (ER) upon synthesis and is trafficked to the Golgi apparatus as the first step of the secretory pathway[1]. Anterograde transport of proteins from the ER to the Golgi is facilitated by coat protein complex II (COPII)-coated membrane carriers. The COPII coat assembles on the cytosolic side of the ER membrane, generating membrane curvature to form coated carriers while specifically recruiting and enveloping newly synthesized cargo proteins[2,3].

COPII comprises five proteins (Sar1, Sec23, Sec24, Sec13 and Sec31) that are essential and highly conserved from yeast to humans[3]. COPII assembly is initiated by the small guanosine triphosphate (GTP) hydrolase Sar1, which inserts its N-terminal amphipathic helix into the outer leaflet of the ER upon nucleotide exchange, an event catalyzed by the ER-resident GTP exchange factor (GEF) Sec12 (refs. [4,5]). Membrane-bound Sar1 recruits heterodimeric Sec23–Sec24 to form

the inner layer of the COPII coat, with Sec24 acting as the main cargo-binding subunit[6,7]. The outer coat layer is formed when heterotetrameric rod-shaped Sec13–Sec31 complexes are recruited to budding sites through the interaction of Sec31 with Sec23–Sar1 and assemble in a cage-like arrangement[8–10]. Symmetric polyhedral cages assemble in vitro when purified Sec13–Sec31 heterotetramers are incubated in the absence of any membrane[10,11]. The detachment of Sar1 from the membrane is triggered by GTP hydrolysis, stimulated by its cognate GTPase-activating protein (GAP) Sec23 and further accelerated by binding of Sec31 (ref. [12]). Sar1 GTP hydrolysis is thought to destabilize the coat; however, the dynamics and regulation of coat disassembly are poorly understood.

We previously determined the structure of the *Saccharomyces cerevisiae* COPII coat reconstituted in vitro from giant unilamellar vesicles (GUVs) using cryo-electron tomography (cryo-ET) with subtomogram averaging (STA)[13–16]. We showed that COPII forms coated tubes on GUVs and that the inner and outer coat layers both arrange

[1]Institute of Structural and Molecular Biology, Birkbeck College, London, UK. [2]Institute of Structural and Molecular Biology, UCL, London, UK. [3]The Francis Crick Institute, London, UK. [4]MRC Laboratory of Molecular Biology, Cambridge, UK. [5]Present address: EMBL, Heidelberg, Germany. [6]Present address: School of Life Sciences, University of Dundee, Dundee, UK. ✉e-mail: g.zanetti@bbk.ac.uk

into pseudohelical lattices that wrap around the tubular membrane. High-resolution STA yielded atomic models describing coat interactions that allowed us to design coat mutants where assembly interfaces are disrupted[15]. We found that the two interfaces that form the outer coat cage, formed by the N-terminal and C-terminal domains of Sec31, are dispensable for membrane budding in vitro and in yeast cells lacking the glycosylphosphatidylinositol-anchored protein cargo adaptor Emp24 (refs. [15,17]). Moreover, when the interface between inner coat lattice subunits was weakened by amino acid substitutions, budded membranes switched from a tubular to a spherical profile, indicating that membrane curvature is generated by a complex network of interactions spanning both coat layers[15].

COPII-coated membrane carriers are known to adopt a range of sizes and shapes, which may be important to adapt to the wide range of cargoes that need to be accommodated. However, it remains unclear how coat assembly is regulated to achieve a variety of membrane carrier sizes[3,18,19]. Whilst our previous studies found that purified *S. cerevisiae* COPII forms extended tubules on GUVs, electron microscopy (EM) studies of cell sections suggested that membrane carriers in vivo are spherical vesicles 50–100 nm wide[20–22], raising the question of which components of native membrane composition affect coat assembly and budding morphology. It also remains unclear how the tightly packed inner coat assembly is compatible with cargo binding by the Sec24 subunits. To answer these questions, we carried out in vitro reconstitution of COPII budding using native ER membranes derived from yeast, referred to as microsomes. In striking comparison to the tubules formed by COPII on GUVs, cryo-ET revealed that the majority of coated membranes are pseudospherical. We used STA[16,23,24] to obtain the structures of the inner and outer coat assembled on native membranes. We found that the inner coat layer can assemble as in its tubular arrangement but forms limited patches of coat that are randomly oriented around a pseudospherical membrane. Cargo density could be detected within the inner coat array, in the space between inner coat subunits, indicating that the lateral assembly of Sar1–Sec23–Sec24 heterotrimers can occur while small or flexible cytosolic domains of cargo molecules are accommodated in between. Lastly, STA analysis of the outer coat layer revealed nonsymmetric cages with a variety of architectures. We characterize multiple points of flexibility, increasing the complexity of the outer coat network and challenging the current model where assembly of the outer coat into a polyhedral cage is the main driver of membrane curvature.

## Results

### COPII forms coated pseudospherical vesicles on microsomes

To reconstitute COPII budding in vitro from native membrane sources, we incubated purified *S. cerevisiae* COPII proteins with *S. cerevisiae* ER-enriched microsomes and a nonhydrolyzable GTP analog (GMP-PNP) (Fig. 1a and Extended Data Fig. 1a). Imaging these budding reactions using cryo-ET revealed that COPII primarily forms vesicles (96.3% of all coated membranes) on microsomal membranes that are clearly coated with both the inner and the outer coat (Fig. 1b). Only a minority of coated tubules were observed (3.7%) in striking comparison to previous reconstitutions using GUVs (91.4% tubules) (EM Public Image Archive (EMPIAR)-11257). Most vesicles appeared fully coated (Fig. 1b and Extended Data Fig. 1b).

The microsome-derived COPII-coated vesicles were significantly smaller than the donor membranes, as measured in a control sample where guanosine diphosphate (GDP) was supplemented in place of GMP-PNP, demonstrating that the membrane is being actively deformed by COPII (Fig. 1c). Most vesicles were detached, with only a handful of instances where coated vesicles were connected to other membranes by a constricted neck (Fig. 1b and Extended Data Fig. 1c). Given that we used nonhydrolyzable GTP analogs and performed no centrifugation or other mechanical perturbation of the sample, this suggests that vesicle scission from donor membranes may not depend on GTP hydrolysis.

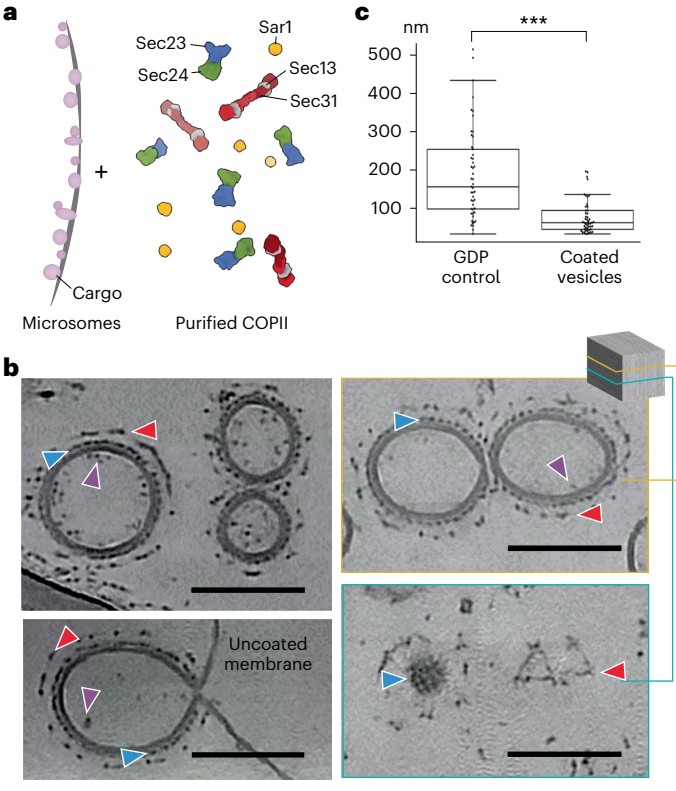

**Fig. 1 | Overview of the in vitro reconstitution experiment. a**, Schematic of the in vitro reconstitution approach. **b**, *XY* slices of representative reconstructed tomograms where instances of the inner coat, outer coat and cargo are labeled with blue, red and purple arrowheads, respectively. Bottom left, an example of a vesicle connected to its origin membrane by a neck. Right, two *Z* slices of the same tomogram. The bottom slice shows both the inner and the outer coat layers of neighboring vesicles. Scale bar, 100 nm. **c**, Membrane diameters were measured from a control reconstitution reaction where GDP was used and compared with the diameters of coated membranes obtained in the presence of GMP-PNP ($n = 1$). Box plots centers, boundaries and whiskers represent the median, the 25th and 75th percentiles and the minimum and maximum values within box boundaries + 1.5× the interquartile range, respectively. Statistical significance was determined using a homoscedastic two-tailed *t*-test ($P = 1.7 \times 10^{-8}$).

### The inner coat lattice assembles in small patches on vesicles

Previous high-resolution STA structures of GUV-derived tubules showed that the inner coat assembles laterally to form a pseudohelical lattice[15,16]. To assess whether and how the previously characterized assembly interfaces can give rise to spherical vesicles, we used STA to obtain a structure of the inner coat on vesicles (Fig. 2a,b and Table 1). We found that the arrangement of a subset of the inner coat is analogous to that previously described on tubes, with Sar1–Sec23–Sec24 trimers assembling laterally and longitudinally in an ordered lattice (Fig. 2a–c and Extended Data Fig. 1d). At the resolution obtained (14.5 Å), there were no noticeable differences in the overall structure of the inner coat between the vesicles and the tubes, aside from the underlying membrane having a spherical rather than tubular curvature. Consequently, we could unambiguously fit a previous high-resolution structure (Protein Data Bank (PDB) 8BSH) of the inner coat into the density. However, the overall arrangement of the inner coat lattice differed greatly. On spherical vesicles, the inner coat lattice formed in small patches (Fig. 2c). These patches could be orientated in different directions to one another on the same vesicle, suggesting that separate inner coat arrays can coexist at multiple sites on the vesicle surface (Fig. 2c). Visual inspection of tomograms showed that vesicles appeared to have an inner coat even where patches were not detected,

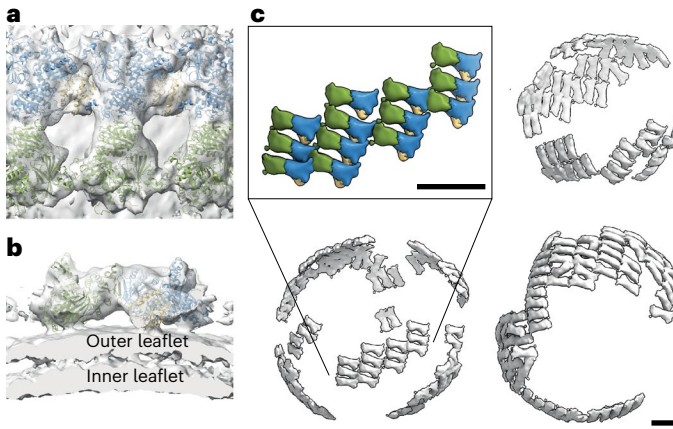

**Fig. 2 | Subtomogram averaging of the COPII inner coat on microsomes.**
**a**,**b**, STA of the inner coat on vesicles fitted with three copies of the Sec23–Sec24–Sar1 heterotrimer atomic model (PDB 8BSH) with Sar1 in yellow, Sec23 in blue and Sec24 in green. Views looking down toward the membrane (**a**; top view) and cutting through the membrane (**b**; side view). **c**, A low-pass-filtered STA structure is mapped back in space. Three examples are shown to demonstrate the small patches arrangement of the lattice. Inset, a close up of one of the patches with the same color code as in **a**. Scale bar, 10 nm.

indicating that ordered patches and unordered individual subunits coexist on spherical membranes (Extended Data Fig. 1d).

## Cargo binds within the COPII inner coat lattice

We next set out to establish whether inner coat lattice formation is compatible with the presence of cargo. The inner coat is known to bind to a range of cargo molecules on several previously characterized binding sites on Sec24, including the A-site located on the Sec24 side distal to Sar1 within the heterotrimer and the B-site, C-site and D-site located closely to one other on the opposite face of Sec24 (refs. 25,26). If cargo is bound to Sec24 in our structure, we would expect to see extra protein density proximal to the known binding sites. As we were unable to visualize density clearly above noise levels, we calculated the difference map between our STA structure of the inner coat on microsomes and a map generated by low-pass filtering the fitted model of the Sar1–Sec23–Sec24 heterotrimer to 14.5 Å. From the difference map, we found strong signal in the space between neighboring Sec24 subunits, indicative of the presence of protein density and, thus, potentially cargo (Fig. 3a). The difference density seemed to be located closest to the B and C cargo-binding sites of Sec24. As a control, we repeated the same experiment using the previously determined structure of the inner coat on cargoless GUVs[16], for which the difference map appeared clear of any density (Fig. 3b).

To further support the hypothesis that the extra density we see is bona fide cargo, we analyzed the composition of microsomal membranes by mass spectrometry (MS). We detected many known Sec24 cargo proteins among the most abundant constituents of these membranes. These include export receptors (Erv25, Erv29, Erv46, Erv41, Emp24, Svp26 and several Erp proteins), components of the Golgi SNAP receptor (SNARE) complex (Sec22, Sed5, Bet1 and Bos1), HDEL receptor (Erd2) and other Sec24 interactors (Yor1, Prm8 and Shr3). We also detected a high number of proteins that are localized to other compartments such as the plasma membrane, Golgi and endosomal network. While these could come from contaminant membranes in the microsome preparation, we assume that a subset belonged to a newly synthesized pool located in the ER and ready to enter COPII vesicles. Cargo proteins and recycling receptors were previously shown to be enriched in COPII-coated vesicles over prevailing concentration in microsomal membranes[26,27].

Because of the presence of a wide range of structurally diverse cargoes on the microsomal membranes, it was not possible to resolve

the bound protein density to anything other than a shapeless 'blob' (Fig. 3a). It is likely that the cargoes bound to the inner coat within the lattice are small and/or flexible, as cargoes with bulky cytosolic domains would be sterically prevented from binding in the 50-Å-wide space between neighboring heterotrimers (Fig. 3a). Whilst we expect different subsets of Sec24 molecules to be bound to cargos of different sizes or not at all, we were unable to reproducibly differentiate between them using three-dimensional (3D) classification. This is likely because of the high amount of compositional and conformational heterogeneity of the cargo molecules and the fact that different sites on Sec24 may be bound substoichiometrically to different cargoes.

To further test whether inner coat lattice formation is compatible with cargo binding, we reconstituted COPII budding using GUVs whose surface was enriched with the cytosolic domain of a small cargo protein, Sed5 from *S. cerevisiae*. Sed5 is a SNARE protein that acts at the *cis*-Golgi in complex with other SNAREs[28]. Sed5 contains two known Sec24-binding motifs specific for the A-site and B-site (YNNSNPF and LMLME, respectively)[25]. Biochemical studies have shown that the YNNSNPF motif on Sed5 is occluded in the monomeric state but becomes exposed in the context of the SNARE complex[25], raising interesting questions about regulation of its transport. An analogous conformational switch was reported for other SNARE proteins of the syntaxin family[29,30]. AlphaFold predictions of the Sed5 structure (AF-Q01590-F1) suggest that both Sec24-binding peptides are found in a highly flexible region characterized by very low confidence scores, allowing Sed5 to bind in the small space between inner coat units (Extended Data Fig. 2a).

First, we purified the Sed5 cytosolic domain (residues 1–319) to high purity and homogeneity (Extended Data Fig. 2b). We enriched the surface of the GUVs with Sed5 by the association of Ni-NTA tagged lipids in the GUVs to a C-terminal 6xHis-tag in the purified Sed5, cloned in place of the transmembrane domain (Fig. 3c, inset). We verified the successful association of Sed5 to the membrane by liposome flotation assays (Extended Data Fig. 2b). We then carried out COPII budding reconstitution in vitro using Sed5-enriched GUVs (Fig. 3c). Imaging these budding reactions using cryo-ET revealed that COPII primarily forms tubes (88.8% of all coated membranes) (Fig. 3d), similarly to previous studies with cargoless GUVs[15]. The inner and outer coat lattices were clearly visible on these tubes (Fig. 3d).

To establish whether Sed5 was bound within the inner coat lattice, we carried out STA to generate a high-resolution (4.1 Å) structure of the inner coat lattice (Extended Data Fig. 3a–c and Table 1). The Sed5-bound map was essentially identical to previous structures lacking cargo but, crucially, we saw unambiguous protein density in one of the known Sed5-binding pockets in correspondence to the B-site (Fig. 3e,f). We were unable to resolve any further Sed5 protein density outside of the known binding pocket on Sec24. This is unsurprising given that the Sec24-binding motifs on Sed5 were predicted to be in a highly flexible and disordered region (Extended Data Fig. 2a). The A-site appeared unoccupied (Extended Data Fig. 3), consistent with previous studies that showed that the A-site-specific YNNSNP peptide is occluded in the monomeric form of Sed5. This peptide only becomes exposed when Sed5 forms SNARE complexes with its partners; thus, monomeric Sed5 is expected to bind the B-site[25]. We confirmed that Sed5, as presumably other small and flexible cargo proteins, can bind to the inner coat without disrupting the lattice. The lattice disruption we observe on vesicles derived from native microsomes is, therefore, probably because of the presence of more bulky proteins. We note that, given the purity of the Sed5 preparation (Extended Data Fig. 3) and the absence of any additional differences with the cargoless budding reaction, it is highly likely that the extra density we see corresponds to Sed5.

## The COPII outer coat is heterogeneous on vesicles

The Sec13–Sec31 outer coat layer was clearly visible on microsome-derived COPII-coated vesicles (Fig. 1b). Manual inspection of denoised tomograms revealed that the outer coat was generally arranged in

**Table 1 | Cryo-EM data collection, refinement and validation statistics**

| | Microsome inner coat (EMD-19417) | Microsome outer coat vertex (EMD-19421) | Microsome outer coat rods (EMD-19418) | Microsome outer coat fifevold (EMD-19879) | Sed5–GUV inner coat (EMD-19410) | Sed5–GUV outer coat vertex (EMD-19414) | Sed5–GUV outer coat rods (EMD-19416) |
|---|---|---|---|---|---|---|---|
| **Data collection and processing** | | | | | | | |
| Magnification | 81,000 | | | | 64,000 | | |
| Voltage (kV) | 300 | | | | 300 | | |
| Total Electron exposure (e⁻ per Å²) | 140 | | | | 142 | | |
| Electron exposure per tilt (e⁻ per Å²) | 3.41 | | | | 3.46 | | |
| Defocus range (µm) | −3 to −5 | | | | −1.5 to −3.5 | | |
| Pixel size (Å) | 1.526 | | | | 1.33 | | |
| Symmetry imposed | None | | | | None | | |
| Energy filter slit width (eV) | 20 | | | | 10 | | |
| Tilt range | −60°, 60° | | | | −60°, 60° | | |
| Tilt increments | 3° | | | | 3° | | |
| Acquisition scheme | Bidirectional | | | | Bidirectional | | |
| Picked particle images (no.) | 33,075 | 23,939 | 25,369 | 461 | 290,658 | 178,700 | 75,404 |
| Final particle images (no.) | 12,187 | 19,368 | 18,852 | 461 | 178,700 | 18,852 | 39,757 |
| Map resolution (Å) FSC threshold | 14 0.143 | 11.4 0.143 | 11.8 0.143 | 34 0.143 | 4.1 0.143 | 9.7 0.143 | 9.5 0.143 |
| Local resolution range (Å) | 13.6–24.4 | 9.7–24.5 | 9.7–24.5 | 24.4–37.2 | 3.5–8.5 | 8.9–17.5 | 7.0–13.5 |

cage-like structures, with 'rods' of Sec13–Sec31 acting as edge elements. Multiple arrangements of these Sec13–Sec31 rods were observed, as shown in some example cages depicted in Fig. 4a. In many instances, four rods converged to form vertices through the interaction of Sec31 N-terminal β-propeller domains, in the canonical manner previously described for in vitro assembled protein-only cages and reconstituted tubules[10,11,15] (Fig. 4a, bottom panel; red spheres). We also observed rods where one or both of the Sec31 N-terminal domains bind to the Sec31 dimerization domain (Fig. 4a, bottom panel; dotted lines). We previously described a similar interaction on tubules and proposed that it stabilizes the outer coat when neighboring patches are 'out of phase' with respect to one another and vertices cannot form[15]. Lastly, we observed a new interaction where five rods converge to form vertices (Fig. 4a, bottom panel; blue spheres).

The edge elements in the cages outline triangular, rhomboidal and pentameric faces (Fig. 4a). These were previously described in in vitro assembled protein-only cages, where regular cuboctahedral and icosidodecahedral structures were formed in high-salt conditions[10,11]. However, in our physiological buffer conditions and while coating native membranes, no such global symmetry was detected.

The structures of the canonical outer coat vertex and Sec13–Sec31 rods were resolved by STA to 11–12 Å for both the microsome-derived and the Sed5-enriched GUV samples (Table 1). This resolution allowed unambiguous rigid-body fitting using previously determined atomic models (PDB 2PM9 and PDB 2PM6) (Fig. 4b,c and Extended Data Fig. 4). For both vertices and rods, the Sed5–GUV and microsome-derived maps were very similar (Extended Data Fig. 4). Previously, we showed that the Sec31 C-terminal domain binds to the dimerization domain of another Sec31 to form an 'elbow' (Fig. 4c, black arrowheads) and hypothesized that this interaction is important to stabilize the coat. However, the microsome-derived structure contained a stronger and better-defined density for the C-terminal domain of Sec31 (Extended Data Fig. 4e,f). Taken together, this suggests a more prominent role for this stabilizing interaction in the context of the widely varying assembly seen on the spherical vesicles derived from microsomes compared with the GUV-derived tubules. We also solved the structure of the vertex formed by convergence of five rods (Fig. 4d). Although the resolution

is low because of the limited number of particles, the shape and size of Sec13–Sec31 rods are clearly recognizable, indicating that these vertices are formed by the interaction of five Sec31 β-propeller domains.

The refined positions and orientations of vertex and edge elements obtained by STA allow us to make a quantitative analysis of cage architecture. When we plot the positions of neighboring vertices for each vertex (Fig. 5a), we notice that neighbors cluster at expected positions, being roughly 300 Å apart and forming an 'X' shape. However, their diffuse 'cloud' distribution clearly indicates a wide range of possible angles formed at vertices, both tangential (Fig. 5a, left) and normal (Fig. 5a, right) to the membrane. We measured the average angle below each vertex (α). We found that it can assume values between 120° and 180° and that these change continuously rather than clustering in 'preferred' angles, as also seen in the neighbor plot. For each vertex, we plotted the α-angle versus vesicle diameter and found a strong positive correlation (Fig. 5b), indicating that the outer coat structure is related to membrane curvature.

Analysis of the arrangement of the outer coat revealed further unexpected heterogeneity. By plotting the positions of the nearest vertex neighbors for all rods (Fig. 5c), a general rhomboidal and triangular pattern emerged, conveying the expected outer coat arrangement of rods with a vertex at each extremity. However, the points representing the positions of the vertices relative to the center of each rod did not form a sharp peak. Instead, the distribution of the distance of the vertices relative to the rods was broad, which suggested that the rods are not rigid. To investigate this further, we defined distinct classes of rods on the basis of the distance to their nearest vertex from within the 'cloud' of points (blue, purple and red masks in Fig. 5c, inset). We reconstructed the corresponding classes of rods, generating three different maps that demonstrated variation in rod structure (Fig. 5d). Specifically, this analysis revealed high mobility around two previously unidentified hinge regions located near the Sec13 and Sec31 β-propellers (Fig. 5d and Supplementary Video 1).

In summary, the variety of arrangements presented here suggests that the outer coat is highly morphologically heterogenous on native vesicles, in stark contrast to the more regular outer coat morphology seen on tubes and on in vitro assembled membraneless polyhedral cages.

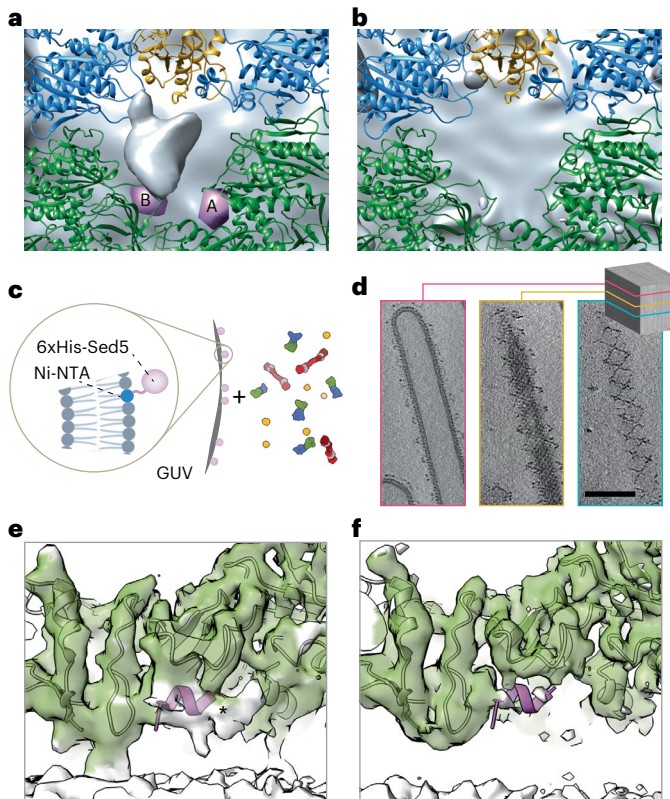

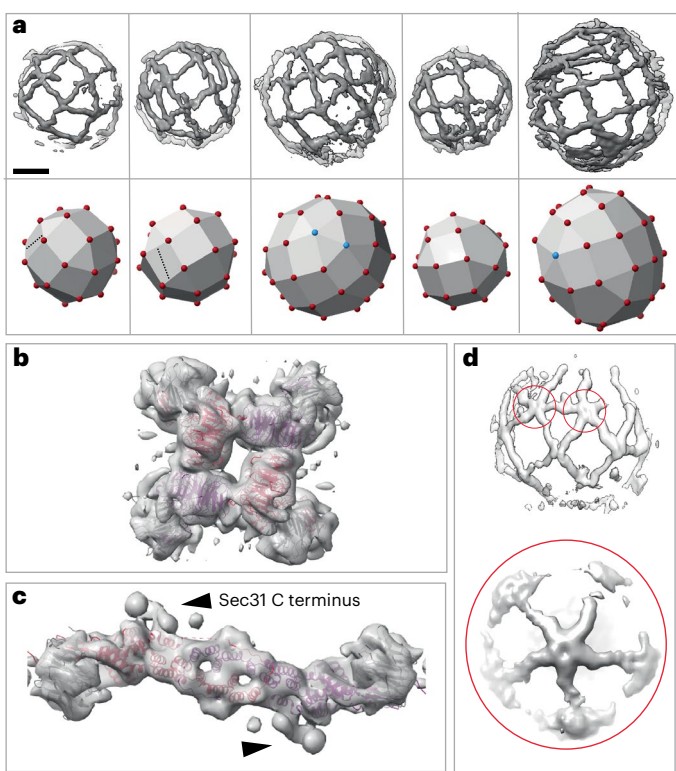

**Fig. 3 | Cargo binding on the COPII inner coat. a**, Difference map between the STA of the inner coat on microsomes and a 14-Å low-pass-filtered volume representation of the fitted model of the inner coat from cargoless GUVs (PDB 8BSH) with Sar1 in yellow, Sec23 in blue and Sec24 in green. The Sec24 A and B cargo-binding sites are represented with pink blobs generated from bound cargo peptides described in previous X-ray crystallography studies (PDB 1PD0 and PDB 1PCX). **b**, As in **a** but using the STA map previously obtained from GUVs (EMD-15949) and low-pass-filtered to 14 Å. **c**, Schematic of the in vitro reconstitution of COPII budding from Sed5-enriched GUVs. Inset, details of 6xHis-tagged Sed5 associating to the Ni-NTA tagged lipids on the GUVs. **d**, *XY* slices through a representative tomogram of Sed5-enriched GUV budding reactions at different *Z* heights displaying the coated tube morphology (pink, *Z* = 162) and the inner (yellow, *Z* = 137) and outer (blue, *Z* = 128) coat arrangements. Scale bar, 100 nm. **e**, Detail of the STA map of Sed5-bound inner coat showing the region around the Sec24 B-site. Density closer than 3.5 Å to the fitted model of the inner coat (PDB 8BSH) is shown in green while white density corresponds to regions of the map that are not explained by the fitted model. The model of a Sec24-bound peptide from Bet1 cargo (PDB 1PCX), which contains the same B-site binding motif as Sed5 (LxxLE), is also fitted to highlight the location of the B-site and is shown in purple. White density in correspondence of this peptide is marked with an asterisk. **f**, As in **e** but displaying the map obtained from cargoless GUV reconstitution (EMD-15949).

**Fig. 4 | Structure and arrangement of the COPII outer coat on microsomes. a**, COPII outer coat arrangement on vesicles. Top panel, outer coat as visible in denoised tomograms. Tomograms were masked for visual clarity. Bottom panel, schematic representation of the cage architecture seen in the top panels. Red and cyan spheres indicate vertices with four and five rods converging, respectively; dotted lines indicate rods forming nonvertex interactions. Scale bar, 30 nm. **b**, STA map of the outer coat vertex on vesicles from microsomes at 11.4 Å, with four copies of the atomic model of the Sec13–Sec31 'vertex element' fitted (PDB 2PM9) with Sec31 in red and purple and Sec13 in gray. **c**, STA map of the outer coat rod on vesiclefs at a resolution of 11.5 Å, with the atomic model of the Sec13–Sec31 'edge' element fitted (PDB 2PM6). Color code as in **b**. **d**, Structure of vertices formed by the convergence of five rods. Top, an example tomogram density depicting two five-way vertices (red circles). Bottom, the STA average from 460 manually picked five-way vertex particles.

## Relationship between inner and outer coat

Lastly, we analyzed the relationship between the inner and outer coat on vesicles. Previous studies established that Sec31 binds to Sec23–Sar1 through a ~300-aa-long flexible linker[25,31].

On coated tubules, we previously found that this flexible linker permits the outer coat to float on top of the inner layer. This allows for the two lattices to adapt to a continuous range of membrane curvatures[15]. We also found that, while there is no fixed alignment between the inner and outer coat lattices on tubules, both layers are rotationally aligned as they both follow a helical pattern around the tubules.

To assess whether the Sec31 flexible linker allows some degree of movement of the outer coat bound to the inner coat on native membranes, we plotted the position of inner coat subunits neighboring each outer coat vertex (Extended Data Fig. 5a). The neighbor positions cluster below vertices at the expected distance but assume a very broad distribution that does not show any pattern. This suggest there is no fixed alignment between the inner and outer coat lattices, similar to what we described for coated tubules[15].

To see whether the Sec31 flexible linker allows for rotation of the outer coat with respect to the inner coat or whether the two lattice layers are locally rotationally aligned as seen on tubules[11], we selected a subset of vertices that all had a neighboring inner coat subunit within a limited region (Extended Data Fig. 5a, yellow mask).

If the two layers were rotationally aligned, we would expect that the average of a subset of vertices with fixed translational relationship to neighboring inner coat subunits would show clear density for both layers. Averaging the selected vertex subset produced a map where no discernible inner coat structure was visible below the vertex, indicating that the two layers are both translationally and rotationally unrelated (Extended Data Fig. 5b). This suggests that the flexible linker allows for full translational and rotational freedom and that there are no other factors binding the two layers together in our in vitro reconstitution.

## Discussion

We reconstituted COPII budding in vitro using *S. cerevisiae* microsomes as native membrane sources. Microsomes are cell-derived membranes,

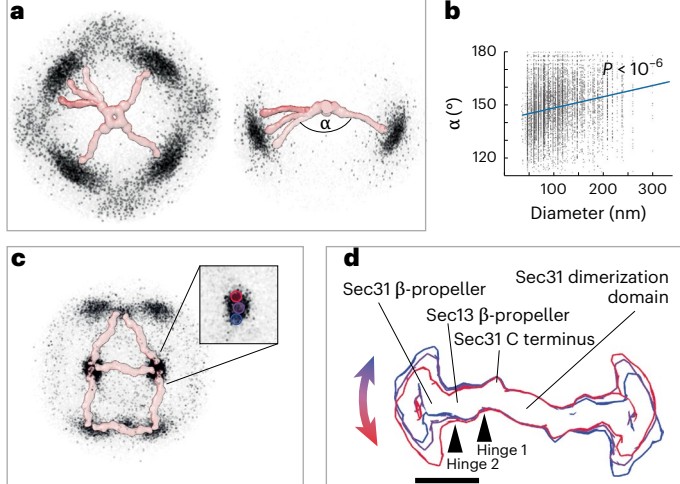

**Fig. 5 | Outer coat flexibility. a**, The position of neighboring vertices surrounding all vertices is plotted; each black dot corresponds to the position of one neighbor. Filtered density maps for vertices and rods are manually overlaid to aid visual interpretation. Left, top view; right, side view. **b**, The average angle below each vertex (as shown in **a**) is plotted against the corresponding vesicle diameter, together with the correlation trendline and *P* value (two-tailed *t*-test, $P = 5.45 \times 10^{-7}$). **c**, The position of vertices surrounding all rods is plotted; each black dot corresponds to the position of one vertex. The pattern appearing from the clustering of nearest vertices corresponds to most rods being arranged in rhomboids and triangles (overlaid rod density). Inset, zoomed-in view of one cluster of neighbor positions, showing masks used to select particle classes. **d**, Variation of rod structures. Rods were selected according to the nearest vertices falling within regions defined by the red, purple and blue masks shown in the inset in **c** and were reconstructed as different classes. The resultant structures are overlaid and show movement around two major hinge regions. Scale bar, 10 nm.

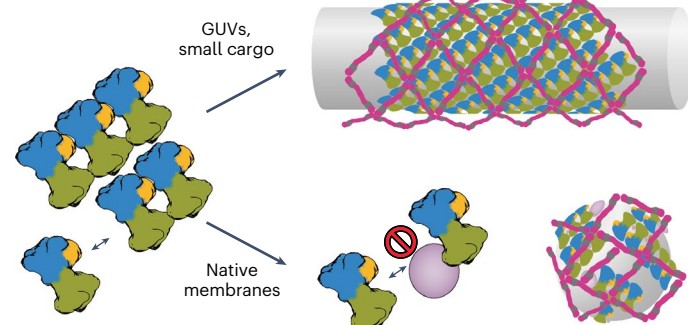

**Fig. 6 | Schematic describing the proposed model of COPII budding.** Inner coat lattice assembly drives curvature of the membrane. If undisturbed, this leads to an extended pseudohelical lattice and formation of coated tubes (top). In native conditions, where bulky proteins (pink) are present, extensive assembly of inner coat is not possible and small patches randomly orient to generate near-spherical membranes (bottom).

purified by sucrose gradient centrifugation, which largely comprise the ER. Therefore, microsomes resemble the ER in their lipid composition, heterogeneity and, importantly, the presence of transmembrane and lumenal cargo proteins. In striking comparison to the COPII-coated tubules generated from GUVs, microsome-derived membrane carriers are mostly pseudospherical. The overall morphology and appearance of the coat reconstituted on microsomes is very reminiscent of the structures seen in situ on cryo-focused ion beam scanning EM data obtained from *Chlamydomonas reinhardtii*[21] and yeast (EMPIAR-11462)[32].

The vast majority of vesicles were detached from the donor membranes, with only a handful of instances of constricted necks. Here, we used a nonhydrolyzable GTP analog, suggesting that GTP hydrolysis is not required for scission in our system, consistent with previous studies[33]. While we did not perform any centrifugation or mechanical perturbation aside from gentle pipetting and blotting to prepare our samples, we cannot exclude scission was triggered by nonphysiological mechanisms. However, similar experiments performed on GUVs with COPII interface mutants resulted in coated spherical profiles that remained linked by constricted necks (like beads on a string)[15], suggesting that scission might depend on factors present within the microsome membrane.

STA analysis of the coat on spherical vesicles showed that the inner coat Sar1–Sec23–Sec24 heterotrimers assemble into small patches of lattice, in contrast to the continuous lattice found on tubules. The arrangements of neighboring subunits in these small patches and in the extended lattice found on regular tubules are highly similar. Cargo protein density was detected within the inner coat lattice of microsome-derived vesicular carriers, indicating that small and/or flexible cargo cytosolic domains can be accommodated within the tightly packed inner coat. This was further confirmed by the reconstitution of the flexible Sed5 cytosolic domain onto GUVs, as COPII budding leads to formation of highly ordered tubules where a short Sed5 peptide can be detected bound to Sec24 B-site. We do not see any notable density for the Sed5 peptide bound to the A-site, consistent with previous descriptions of occlusion of the A-site interaction motif in the monomeric SNARE[25] (Extended Data Fig. 3d,e).

STA analysis of the coat assembled around spherical vesicles also revealed a highly variable and flexible outer coat cage consisting of Sec13–Sec31 rods assembling with many different geometries. Most rods converge through interactions between Sec31 N termini to form canonical four-way vertices but we also detected T-junctions with the dimerization domains of other rods, as well as pentameric vertices. Outer coat cages have elements of icosidodecahedral and cuboctahedral geometry, such as triangular, rhomboidal and pentameric faces, reminiscent of membraneless in vitro assembled cages. However, individual cages are distinct and no overall symmetry is present. The rods themselves are highly flexible, with two major hinge points around the Sec13 β-propeller.

The interaction between the inner and outer coat layers, known to be mediated by a disordered region of Sec31, is also variable with local inner and outer coat lattices being translationally and rotationally not aligned.

Overall, our findings that COPII morphology differs between microsomes and naked GUVs, in combination with our previous finding that the regions responsible for outer coat assembly are not necessary for budding[15], challenge the idea that the outer coat cage assembly is the main driver of membrane curvature. We propose a model for the generation of membrane curvature by the COPII coat (Fig. 6), where vesicle shape is mostly determined by inner coat assembly. According to this model, the extent of inner coat lattice polymerization drives membrane curvature. In the case of undisturbed lattice assembly, as obtained with GUVs in vitro where GTP hydrolysis is inhibited, no bulky proteins are present and membrane sources are abundant, coated tubes are formed. In native conditions, where bulky proteins are present to disrupt inner coat lattice assembly, small patches of randomly oriented inner coat lattice lead to the formation of pseudospherical vesicles. In this scenario, the outer coat's ability to adapt to a continuous and varied range of growing curvature ensures effective binding and assembly of cages, which stabilize the coated vesicle. Of note, in vitro reconstitutions from GUVs using COPII mutants with weakened inner coat lattice interfaces also led to the formation of spherical profiles[15], supporting the proposed model.

## Online content

Any methods, additional references, Nature Portfolio reporting summaries, source data, extended data, supplementary information,

acknowledgements, peer review information; details of author contributions and competing interests; and statements of data and code availability are available at https://doi.org/10.1038/s41594-024-01413-4.

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

## Methods

### Cloning

Sed5 (UniProt Q01590) (residues 1–319, truncating the transmembrane helix) was cloned from the *S. cerevisiae* S288c genome into a pETM-11 expression vector linearized at the XhoI and XbaI restriction sites using In-Fusion (Takara) technology. A flexible triple-glycine linker was added between the C-terminal residue (319) of Sed5 and a 6xHis-tag. The primers used were as follows:

Forward, 5′-GGTGTCCTCCTCCTCTATTACTCTTTATCCTGTC-GAAG-3′

Reverse, 5′-GAGGAGGAGGACACCACCACCACCACCAC-3′

Sec23–Sec24, Sec13–Sec31 and Sar1 constructs previously described by Hutchings et al. (2021)[15] were used here.

### Protein expression and purification

The Sed5 pETM-11 vector was transformed into *Escherichia coli* (BL21) cells by heat shock. Cells were cultured at 37 °C with 220 rpm shaking in 2 L of LB medium supplemented with kanamycin. When cultures reached an optical density between 0.7 and 1, 0.2 mM IPTG was added and the incubation temperature was reduced to 16 °C. Culture pellets were harvested after approximately 16 h by centrifugation and flash-frozen in liquid nitrogen before storage at −80 °C.

Sed5 pellets from 2 L of culture were thawed and resuspended in 20 ml of Ni-A buffer (50 mM Tris (pH 8), 500 mM KCl, 0.1% Tween-20 (v/v), 10 mM imidazole and 1 mM DTT) supplemented with one complete protease inhibitor tablet (Roche). Then, 40 mg ml⁻¹ lysozyme was added and cells were stirred on ice for 20 min. Cells were lysed using a cell disruptor. Unbroken cells were removed by ultracentrifugation at 20,200$g$ for 25 min. The supernatant was loaded onto a Ni-NTA 5-ml His-trap column (GE Biosciences) equilibrated with Ni-A buffer and washed with five column volumes of Ni-A buffer. Sed5 was eluted from the column by applying a linear gradient of Ni-B buffer (50 mM Tris (pH 8), 500 mM KCl, 0.1% Tween-20 (v/v), 500 mM imidazole and 1 mM DTT). Fractions were analyzed by SDS–PAGE and those containing Sed5 were pooled before tenfold dilution in Q-A buffer (20 mM Tris (pH 8.0), 0.1% Tween-20 (v/v), 10% glycerol (v/v) and 1 mM DTT). Sed5 was loaded onto a 5-ml HiTrap Q column (GE Biosciences) equilibrated with Q-A buffer. The column was washed with two column volumes of Q-A buffer and two column volumes of a mixture of 90% Q-A buffer and 10% Q-B buffer (20 mM Tris (pH 8.0), 0.1% Tween-20 (v/v), 10% glycerol (v/v), 1 mM DTT and 1 M KCl). Sed5 was eluted with a linear gradient of Q-B buffer. Fractions were analyzed by SDS–PAGE and those containing Sed5 were pooled and concentrated using a protein concentrator with a 10-kDa molecular weight cutoff to a final concentration of 0.5 mg ml⁻¹. Sed5 was separated into 100-μl aliquots and flash-frozen.

The final step of Sed5 purification was carried out on the day of use. One aliquot of Sed5 was thawed before injection onto a Superdex 200 Increase 3.2/300 column equilibrated with HKM buffer (20 mM HEPES, 50 mM potassium acetate and 1.2 mM MgCl₂, pH 6.8). Fractions containing Sed5 were identified by SDS–PAGE and pooled together.

The purified protein was confirmed as Sed5 by analysis with SDS–PAGE combined with gel sequencing by MS at the MS and Proteomics Facility at the University of St. Andrews.

Sec23–Sec24, Sec13–Sec31 and Sar1 were expressed and purified as described previously from SF9 and *E. coli* cells, including the steps to cleave the 6xHis-tags in Sec23–Sec24 and Sec13–Sec31 (ref. 15).

### Liposome flotation assays

Liposomes were generated as previously described[34] using the 'major–minor' lipid mixture: 49 mol.% phosphatidylcholine, 20 mol.% phosphatidylethanolamine, 8 mol.% phosphatidylserine, 5 mol.% phosphatidic acid, 9 mol.% phosphatidylinositol, 2.2 mol.% phosphatidylinositol-4-phosphate, 0.8 mol.% phosphatidylinositol-4,5-bisphosphate, 2 mol.% cytidine diphosphate–diacylglycerol, supplemented with 2 mol.% Texas red–phosphatidylethanolamine, 2 mol.% Ni-NTA-tagged lipids (18:1 DGS–NTA(Ni)) and 20% (w/w) ergosterol.

Liposomes were premixed with Sed5 and floatation assay experiments were performed without and with the addition of COPII components: 1 μM Sar1, 180 nM Sec23–Sec24, 173 μM Sec13–Sec31, 360 nM Sed5 with 1 mM GMP-PNP (Sigma-Aldrich) and 2.5 mM EDTA (pH 8.0). All flotation assays contained 0.27 mM liposomes in a total volume of 75 μl. Liposome flotation reactions were mixed with 250 μl of 1.2 M sucrose in HKM buffer in an ultracentrifuge tube. Next, 320 μl of 0.75 M sucrose in HKM was gently layered on top. A final layer of 20 μl of HKM was then layered on top of the sucrose solutions. Ultracentrifuge tubes were loaded into a SW-55 Ti ultracentrifuge rotor before spinning at 280,000$g$ at 4 °C for at least 16 h. The top 20 μl of the sucrose gradient was carefully extracted before analysis by SDS–PAGE with silver staining.

### Budding reactions

Purified microsomes from *S. cerevisiae* were prepared as described previously[31]. Then, 1.5 mg of microsomes were washed three times carrying out the following steps: resuspending the microsomes in 1 ml of B88 buffer (20 mM HEPES (pH 6.8), 150 mM potassium acetate, 250 mM sorbitol and 5 mM magnesium acetate), pelleting membranes by centrifugation on a chilled benchtop centrifuge at 20,000$g$ for 2 min, removing the supernatant and resuspending the pellet in 50 μl of B88 buffer. After washing, the pellets were diluted a further eight times and chilled on ice before use in budding reactions.

Budding reactions in microsomes were prepared by incubating 1 μM Sar1, 180 nM Sec23–Sec24, 173 μM Sec13–Sec31 with 1 mM GMP-PNP (Sigma-Aldrich), 2.5 mM EDTA (pH 8.0) and 10% microsomes (v/v).

GUVs were prepared by electroformation[35] from 10 mg ml⁻¹ of a major–minor lipid mixture with 2 mol.% Ni-NTA tagged lipids (described above) in a 2:1 chloroform–methanol solvent mixture, as described previously[14,36]. The lipid mixture was spread over two indium tin oxide-coated glass slides. Then, 300 mM sucrose was suspended in a silicon O-ring between these glass slides and GUVs were generated using a NanIon Vesicle Prep Pro. GUVs in the sucrose solution were added to 500 μl of 300 mM glucose and left to sediment overnight at 4 °C. The supernatant was discarded, leaving a 50-μl pellet of GUVs.

Budding reactions in GUVs with Sed5 were prepared by incubating 1 μM Sar1, 180 nM Sec23–Sec24, 173 μM Sec13–Sec31, 360 nM Sed5 with 1 mM GMP-PNP (Sigma-Aldrich), 2.5 mM EDTA (pH 8.0) and 10% GUVs (v/v). GUVs were premixed with the Sed5 before addition to the COPII components. Budding reactions were incubated for at least 30 min before vitrification for cryo-ET.

### Cryo-ET sample preparation

First, 5 nm BSA-blocked gold nanoparticles (BBI Solutions) were added to the budding reactions at a concentration of 10% (v/v). Then, 4 μl of budding reactions from GUVs or microsomes were added to glow-discharged Lacey carbon films on 300-mesh copper grids (Agar Scientific) and incubated for 60 s, before backblotting on a Leica-GP2 plunge-freezer in 95% humidity with a 4-s blotting time. Vitrified grids were stored in liquid nitrogen before data collection.

### Cryo-ET data collection

Budding reactions with microsomes were imaged using cryo-ET at the European Molecular Biology Laboratory (EMBL) Imaging Center in Heidelberg on a Titan Krios microscope (Thermo Fisher Scientific) operated at 300 kV. The microscope was equipped with a SelectrisX energy filter (Thermo Fisher Scientific) and a Falcon 4 detector (Thermo Fisher Scientific) in counting mode. The pixel size was 1.526 Å and tilt series were taken with a defocus range of −3 μm to −5 μm. Tilt series were acquired using a dose-symmetric tilt scheme[37] over a total exposure of 140 e⁻ per Å² with tilt angles ranging between −60° and +60° with 3° increments. Data collection was controlled using SerialEM[38] and

implementing PACE-tomo[39]. A total of 765 high-quality tilt series were collected, yielding the same number of tomograms.

Budding reactions with GUVs and Sed5 were imaged using cryo-ET at the EMBL Imaging Center in Heidelberg over two sessions of data collection on a Titan Krios microscope operated at 300 kV. The microscope was equipped with a K3 (Gatan) detector and energy filter. The first session was collected in super-resolution mode and the second session was collected in counting mode. Pixel size was 1.33 Å and tilt series were taken with a defocus range of −1.5 μm to −3.5 μm. Tilt series were acquired using a dose-symmetric tilt scheme[37] over a total exposure of 142 e⁻ per Å² with tilt angles ranging between −60° and +60° with 3° increments. Data collection was controlled using SerialEM. A total of 326 high-quality tilt series were collected, yielding the same number of tomograms.

Grids were screened and optimized at the Institute of Structural and Molecular Biology (ISMB) EM facility at Birkbeck College.

## Cryo-ET data processing

The microsome dataset was processed using an alpha-phase development version of RELION 5.0 (4.1-alpha-1-commit-d2053c)[40]. Initially, .mdoc files were renamed as TS_[number]-style to ensure compatibility between RELION and Dynamo scripts used later in the processing workflow. Raw data were then imported into RELION 5.0. Individual tilt movies were motion-corrected and averaged using whole-frame alignment in the RELION implementation of MotionCor2 (refs. 41,42). Contrast transfer fraction (CTF) estimation was carried out using CTFFIND-4.1 (ref. 43) with a defocus range of −25,000 to −55,000 Å and a maximum CTF resolution of 20 Å. Tilt series were manually inspected and poor tilt images were removed using a Napari plug-in (https://github.com/napari/napari/blob/main/CITATION.cff) provided as part of the 'exclude tilt images' job type in RELION 5.0. Tilt series were automatically aligned using the IMOD wrapper for fiducial-based alignment in RELION with a fiducial diameter of 8 nm. Tomograms were reconstructed in RELION at a pixel size of 12.208 Å for visual inspection and particle picking. Tomograms were denoised and missing wedge-corrected using IsoNet for use in manual particle picking[44]. We also generated eight-binned CTF-corrected tomograms for use in PyTOM template matching[45], using IMOD's 'etomo' (ref. 46) function on the IMOD metadata generated by the 'align tilt series' job type in RELION.

The Sed5−GUV dataset was processed using the RELION4_Tomo_Robot (https://github.com/EuanPyle/relion4_tomo_robot/blob/master/CITATION.cff). Individual tilt videos were motion-corrected and averaged using whole-frame alignment with MotionCor2 (ref. 41). Videos collected in super-resolution mode were binned twice during motion correction. Tilt series were created from individual tilt images using IMOD's 'newstack' function. Tilt series were manually inspected using IMOD's '3dmod' visualization function and bad tilts were removed using IMOD's 'excludeviews' function. Tilt series were automatically aligned using Dynamo's automated fiducial-based alignment in the RELION4_Tomo_Robot's 'fast_mode' with a fiducial diameter of 5 nm (ref. 47). CTF estimation was carried out using CTFFIND-4.1 (ref. 43). The dataset was then imported into RELION 4.0 (ref. 16). Tomograms were reconstructed in RELION at a pixel size of 10.64 Å for visual inspection and particle picking. Tomograms were denoised and missing wedge-corrected using IsoNet for visual inspection[44].

## STA

**Microsome dataset.** *Inner coat*. The surface of vesicles in IsoNet-denoised tomograms was defined and segmented using the 'pick particle' plug-in in Chimera as described previously[48,49]. The coordinates of the vesicle surface were used to mask the tomograms to enable manual particle picking in UCSF Chimera, which were assigned Euler angles normal to the membrane. A total of 3,579 particles were extracted in 48 voxel boxes from RELION-reconstructed (nondenoised) tomograms at a voxel size of 9.156 Å. Particles were

assigned random in-plane rotation angles and were averaged to create a reference using Dynamo[47] with 4,697 particles. Particles were then aligned and averaged in Dynamo with the following conditions: a cone range of 10° was applied while 360° in-plane rotation was allowed; particle translation was limited to one voxel in all directions because of the accuracy of the coordinates of the manually picked particles; a C2 symmetry was applied because of the pseudosymmetry of the inner coat at low resolution; a mask covering the area of one inner coat subunit was applied; alignment was carried out for 100 iterations. The resulting Dynamo table was converted to a .star file using 'dynamo2relion' (https://github.com/EuanPyle/dynamo2relion). Particles were imported into an alpha-phase development version of RELION 5.0 (4.1-alpha-1-commit-d2053c) and extracted as pseudosubtomograms at bin 4 (voxel size of 6.104 Å) in 64 voxel boxes. Pseudosubtomograms were generated from the raw tilt series and did not use denoised tomograms. A reference was reconstructed at the same box and voxel size using the 'tomo reconstruct particle' job type. Particles were refined using Refine3D with the reference low-pass-filtered to 30 Å, no mask applied, a particle diameter of 200 Å and all Euler angles limited to local refinements of approximately 9° using the additional argument '--sigma_ang 3'. Poorly aligned particles were removed by 3D classification without particle alignment, no mask applied, six classes and a regularization parameter (T value) of 0.2. A reference was reconstructed at bin 1 in a box size of 196 voxels before the tilt series alignment for each tomogram was refined using 'tomo frame alignment' without fitting per-particle motion or deformations. Particles were re-extracted as pseudosubtomograms at bin 4 and refined as before using a mask over one inner coat subunit.

The structure generated by RELION was used to pick more particles in CTF-corrected (nondenoised) tomograms with PyTOM template matching[45] with dose weighting and CTF correction applied. The template used was filtered to 25 Å.

Coordinates from PyTOM (29,496 particles from 475 tomograms) were imported into RELION 5.0. To remove junk particles, 3D classification was carried out with alignment using restricted tilt and psi Euler angles ('--sigma_rot 3 --sigma_psi 3') but leaving in-plane rotation free, a mask over one inner coat unit and over part of neighboring subunits, the map from the refined manually picked particle low-pass-filtered to 25 Å as a reference, four classes, a T value of 0.1 and a particle diameter of 330 Å. Particles clearly resembling the COPII inner coat were kept and refined under similar conditions to the preceding 3D classification. Particles were cleaned again using 3D classification without alignment with six classes and a T value of 0.2. The resulting particles .star file was merged with the manually picked particles generated earlier. Duplicate coordinates were deleted. Particles were exported to a Dynamo table using 'relion2dynamo' (https://github.com/EuanPyle/relion2dynamo) and were cleaned by neighbor analysis, as previously described[48]. Coordinates were converted back to a .star file using 'dynamo2relion' and reimported into RELION. Particles were refined as before but at bin 2 with all Euler angles limited to local refinements using '--sigma_ang 3'. One more round of tomo frame alignment, with per-particle motion, was carried out before tomo CTF refinement. A final refinement was carried out at bin 2 from 12,187 particles (from 352 tomograms), with limited Euler angles using '--sigma_ang 1.5'. The resolution according to Fourier shell correlation (FSC = 0.143) was 14.4 Å (Extended Data Fig. 6a).

A difference map, as described in Fig. 3a, between this structure and the inner coat from cargoless GUVs was generated. First, a model of the inner coat from cargoless GUVs (PDB 8BSH) was fitted into the inner coat map from microsomes. A volume representation of the fitted model was generated using 'molmap' in UCSF Chimera[50] at high resolution (2 Å) before low-pass filtering to 14 Å in MATLAB. All maps were normalized to the same mean and s.d. before the map from the fitted model was subtracted from our map from microsomes. Another difference map, as described in Fig. 3b, was generated in the same way but using a 14-Å low-pass-filtered electron density map (EMD-15949)

corresponding to the fitted PDB model (PDB 8BSH) instead of our map of the inner coat derived from microsomes.

*Outer coat (vertex)*. Outer coat vertices were manually picked in 30 tomograms. Particles were assigned Euler angles normal to their nearest membrane. Particles were extracted in 64 voxel boxes from RELION-reconstructed tomograms at a voxel size of 12.208 Å (bin 8). Particles were averaged as before for the inner coat to form an initial average. Particles were then aligned and averaged in Dynamo as for the inner coat but with a translational shift of four voxels allowed and with a mask covering the vertex. The resulting map was used as a template to pick more particles in CTF-corrected tomograms with PyTOM template matching on all tomograms[45], as for the inner coat. Particles were cleaned on the basis of their proximity to the membrane of the vesicles. Particles were aligned in Dynamo again and the resulting Dynamo table was converted to a .star file using 'dynamo2relion'.

Vertex particles were imported into RELION 5.0 and extracted as pseudosubtomograms in a box size of 64 voxels and at a voxel size of 6.104 Å (bin 4). Particles were cleaned using 3D classification with refinement restricting the tilt and psi Euler angles ('--sigma_rot 4 --sigma_psi 4') but leaving in-plane rotation free. The 3D classification used three classes, a $T$ value of 0.25 and a particle diameter of 600 Å. Particles containing the vertex were then refined under the same conditions used in 3D classification. Particles were extracted at bin 2 and further refined. Tomo frame alignment, tomo CTF refinement and subsequent refinement at bin 2 (voxel size of 3.052 Å) was iteratively repeated until resolution improvements stopped. The final map was reconstructed from 19,368 particles and had a resolution of 11.4 Å (FSC = 0.143) (Extended Data Fig. 6b).

*Outer coat (rod)*. Outer coat rods were manually picked in all tomograms. Particles were assigned Euler angles normal to their nearest membrane. Particles were extracted from RELION-reconstructed tomograms at bin 8 (voxel size of 12.208 Å), averaged to form a reference and aligned in Dynamo, as per the outer coat vertices. Rods of different length were selected and isolated using neighbor analysis. The resulting Dynamo table was converted to a .star file using 'dynamo2relion'.

Rod particles were imported into RELION 5.0 and extracted as pseudosubtomograms in a box size of 64 voxels at bin 8. As for the outer coat vertices, particles were progressively unbinned from bin 8 to bin 2 (voxel size of 3.052 Å) and refined with restrictions to apply local Euler angle searches. The final map was reconstructed from 18,852 particles and had a resolution of 11.8 Å (FSC = 0.143) (Extended Data Fig. 6c).

To produce the maps in Fig. 5d, particles were separated into different groups depending on the position of the neighboring vertices as defined by masks on the neighbor plot. Classes contained between 1,500 and 2,500 particles each, were binned eight times and were not filtered beyond their Nyquist (at 24 Å).

*Outer coat (vertex, five-way rods)*. We manually picked vertices formed by the convergence of five rods (as judged from visual inspection; Fig. 4A). We extracted particles ($n$ = 461) from IsoNet-corrected tomograms at bin 8 (voxel size of 12.208 Å) in 64 voxel boxes, assigned initial angles normal to the membrane and randomized the in-plane rotation, before averaging them to obtain a starting reference for alignments. We used Dynamo to align particles by restraining the angles normal to the membrane within a 20° cone and allowing full searches for in-plane rotation. After 50 iterations, alignments had converged. We imported the aligned coordinates in RELION 4.1 and ran a refinement at bin 8 using angular restraints ('--sigma_ang 3'). The final map was reconstructed from 461 particles and had a resolution of 34 Å (FSC = 0.143).

**Sed5–GUV dataset.** *Inner coat*. The surface of tubes in RELION-reconstructed tomograms was defined and segmented using the 'pick

particle' plug-in in Chimera as described previously[48,49]. The surface of the tube was oversampled and coordinates were assigned Euler angles normal to the membrane. Particles were extracted in 32 voxel boxes from RELION-reconstructed tomograms at a voxel size of 10.8 Å. Particles were then aligned and averaged in Dynamo as before for the microsome inner coat dataset with several differences: in-plane rotation was restricted to 20° with azimuth flipping enabled; $C1$ symmetry was applied; particle translation was limited 15 voxels in all directions; alignment was carried out for one iteration. Duplicates defined as particles within four voxels of another particle and were deleted with Dynamo's 'separation in tomogram' function during alignment. A previous inner coat structure (EMD-11199)[16] was low-pass filtered and used as a reference. Particles were cleaned by neighbor analysis as before for the microsome inner coat dataset. The resulting Dynamo table was converted to a .star file using 'dynamo2relion'.

Particles were imported into RELION 5.0 and extracted as pseudosubtomograms at bin 8. They were refined and progressively unbinned iteratively until bin 1 before tomo frame refinement and tomo CTF refinement as previously described[16]. The final map was reconstructed from 178,700 particles and had a resolution of 4.1 Å (FSC = 0.143) (Extended Data Fig. 6d). The map was sharpened using RELION's LocalRes sharpening with a −50 $B$ factor.

*Outer coat (vertex)*. To pick outer coat vertices, we used the refined coordinates for the inner coat lattice and radially shifted them away from the membrane by 12 pixels. This was done to randomly oversample outer coat subunits at the expected radial distance from the tubular membrane. We then extracted these particles in a 64 voxel box size from RELION-reconstructed tomograms using Dynamo before aligning to a low-pass filtered of a previous vertex structure (EMD-11194)[15]. Alignment parameters were the same as for the inner coat alignment from the Sed5–GUV dataset except $C2$ symmetry was applied. The resulting Dynamo table was converted to a .star file using 'dynamo2relion'.

Vertex particles were imported into RELION 5.0 and extracted as pseudosubtomograms in a box size of 128 voxels at bin 4. As for the outer coat vertices from microsomes, particles were progressively unbinned from bin 8 to bin 2 and refined with restrictions to apply local Euler angle searches. The final map was reconstructed from 13,529 particles and had a resolution of 9.7 Å (FSC = 0.143) (Extended Data Fig. 6e). The map was sharpened using RELION's LocalRes sharpening with a −175 $B$ factor.

*Outer coat (rod)*. To pick outer coat rods, we used the refined coordinates of the outer coat vertices and used Dynamo's subboxing function to create four new coordinates where the rods are placed relative to each vertex. As before for the outer coat vertices, particles were aligned in Dynamo to a low-pass-filtered reference (EMD-11193)[15]. Particles were cleaned by neighbor analysis and duplicates were deleted. The resulting Dynamo table was converted to a .star file using 'dynamo2relion'.

Rod particles were imported into RELION 5.0 and extracted as pseudosubtomograms in a box size of 128 voxels at bin 4. As for the outer coat rods from microsomes, particles were progressively unbinned from bin 8 to bin 2 and refined with restrictions to apply local Euler angle searches. The final map was reconstructed from 39,757 particles and had a resolution of 9.5 Å (FSC = 0.143) (Extended Data Fig. 6f).

In all cases, relevant atomic model coordinates were rigid-body fitted into our maps using UCSF Chimera or ChimeraX. In all cases, fitting was unambiguous.

The number of particles for each dataset is summarized in Table 1. Local resolution was estimated using RELION 5.0 locres implementation.

## MS analysis

Total protein from the *S. cerevisiae* ER microsomes used in the reconstitution experiments ($n$ = 1) was digested using the SP3 method[51]

Article

with some adaptations. In brief, after reduction and alkylation of cysteines, total protein was precipitated onto magnetic beads (MagResyn Hydroxyl, Resyn Biosciences) by adding ethanol to a final concentration of 80% (v/v). Digestion was carried out by incubating the washed magnetic beads and total protein aggregated material with 1 µg of trypsin (Promega) dissolved in 25 mM ammonium bicarbonate containing 0.1% RapiGest detergent (Waters). The sample was then acidified with trifluoroacetic acid to a final concentration of 0.5% (v/v) to stop digestion and induce RapiGest degradation. Magnetic beads and RapiGest-insoluble degradation products were pelleted by centrifugation at 11,000$g$ for 15 min and the supernatant containing tryptic peptides was then taken for MS analysis.

Liquid chromatography (LC)–MS/MS was performed on an Ultimate U3000 high-performance LC system (Thermo Fisher Scientific) hyphenated to an Orbitrap QExactive Classic MS instrument (Thermo Fisher Scientific). Peptides were trapped on a C18 Acclaim PepMap 100 (5 µm, 300 µm × 5 mm) trap column (Thermo Fisher Scientific) and eluted onto a C18 Easy Spray Column (2 µm, 75 µm × 500 mm; Thermo Fisher Scientific) using 180-min gradient of acetonitrile (5–40%). For data-dependent acquisition, MS1 scans were acquired at a resolution of 70,000 (automatic gain control (AGC) target of $1 × 10^6$ ions with a maximum injection time of 65 ms) followed by ten MS2 scans acquired at a resolution of 17,500 (AGC target of $2 × 10^5$ ions with a maximum injection time of 100 ms) using a collision-induced dissociation energy of 25. Dynamic exclusion of fragmented $m/z$ values was set to 40 s.

Raw data were imported and processed in Proteome Discoverer version 3.1 (Thermo Fisher Scientific). The raw files were submitted to a database search using Proteome Discoverer with Sequest HT against the UniProt reference proteome for *S. cerevisiae*. The processing step consisted of a double iterative search using the INFERIS rescoring algorithm on a first pass with methionine oxidation and cysteine carbamidomethylation set as variable and fixed modifications, respectively. For the second pass, all spectra with a confidence filter worse than 'high' were researched with Sequest HT including additional common protein variable modifications (deamidation (N,Q), Q to pyro-E (Q), N-terminal acetylation and methionine loss). The spectral identification was performed with the following parameters: MS accuracy, 10 ppm; MS/MS accuracy of 0.02 Da; up to two trypsin missed cleavage sites allowed. The percolator node was used for false discovery rate (FDR) estimation and only rank 1 peptide identifications of high confidence (FDR < 1%) were accepted.

#### Reporting summary

Further information on research design is available in the Nature Portfolio Reporting Summary linked to this article.

## Data availability

Data supporting the findings of this paper are available from the corresponding author upon reasonable request. The EM maps and models were deposited to the EM Data Bank with the following accession codes: COPII inner coat on microsome vesicles, EMD-19417; COPII outer coat (rod) (long) on microsome vesicles, EMD-19421; COPII outer coat (vertex) on microsome vesicles, EMD-19418; COPII outer coat (five-way vertex) on microsome vesicles, EMD-19879; COPII inner coat on tubes from Sed5–GUVs, EMD-19410; COPII outer coat (rod) on tubes from Sed5–GUVs, EMD-19414; COPII outer coat (vertex) on tubes from Sed5–GUVs, EMD-19416. Source data are provided with this paper.

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

## Acknowledgements

We thank N. Lukoyanova and S. Chen at the ISMB Birkbeck Cryo-EM Lab, Z. Yang and W. Hagen at the EMBL Imaging Center in Heidelberg and eBIC for cryo-ET data collection, D. Houldershaw at Birkbeck College for computational support and K. Downes at the Francis Crick Institute for help with the figures. We thank lab members S. van der Verren and K. Downes for comments on the paper. We acknowledge the access and services provided by the Imaging Center at the EMBL, generously supported by the Boehringer Ingelheim Foundation. We acknowledge the ISMB EM facility (Birkbeck College, University of London), supported by the Wellcome Trust (202679/Z/16/Z and 206166/Z/17/Z). We thank F. Begum for help with MS analysis, which was performed at the Biological Mass Spectrometry and Proteomics Facility of the Medical Research Council (MRC) Laboratory of Molecular Biology (LMB). This work was supported by grants to G.Z. from the European Research Council (ERC-StG-2019 grant 852915) and the Biotechnology and Biological Sciences Research Council (BBSRC

grant BB/T002670/1). Work in the group of E.A.M. was supported by the MRC, as part of UK Research and Innovation (MC_UP_1201/10).

## Author contributions

Conceptualization, G.Z.; funding acquisition, G.Z.; sample preparation for cryo-EM, E.P.; cryo-ET data collection, E.P.; cryo-ET and STA data processing, E.P. and G.Z.; microsome preparation and MS, E.A.M.; writing (original draft), E.P. and G.Z.; writing (revisions), E.P., E.A.M. and G.Z.

## Competing interests

The authors declare no competing interests.

## Additional information

**Extended data** is available for this paper at https://doi.org/10.1038/s41594-024-01413-4.

**Correspondence and requests for materials** should be addressed to Giulia Zanetti.

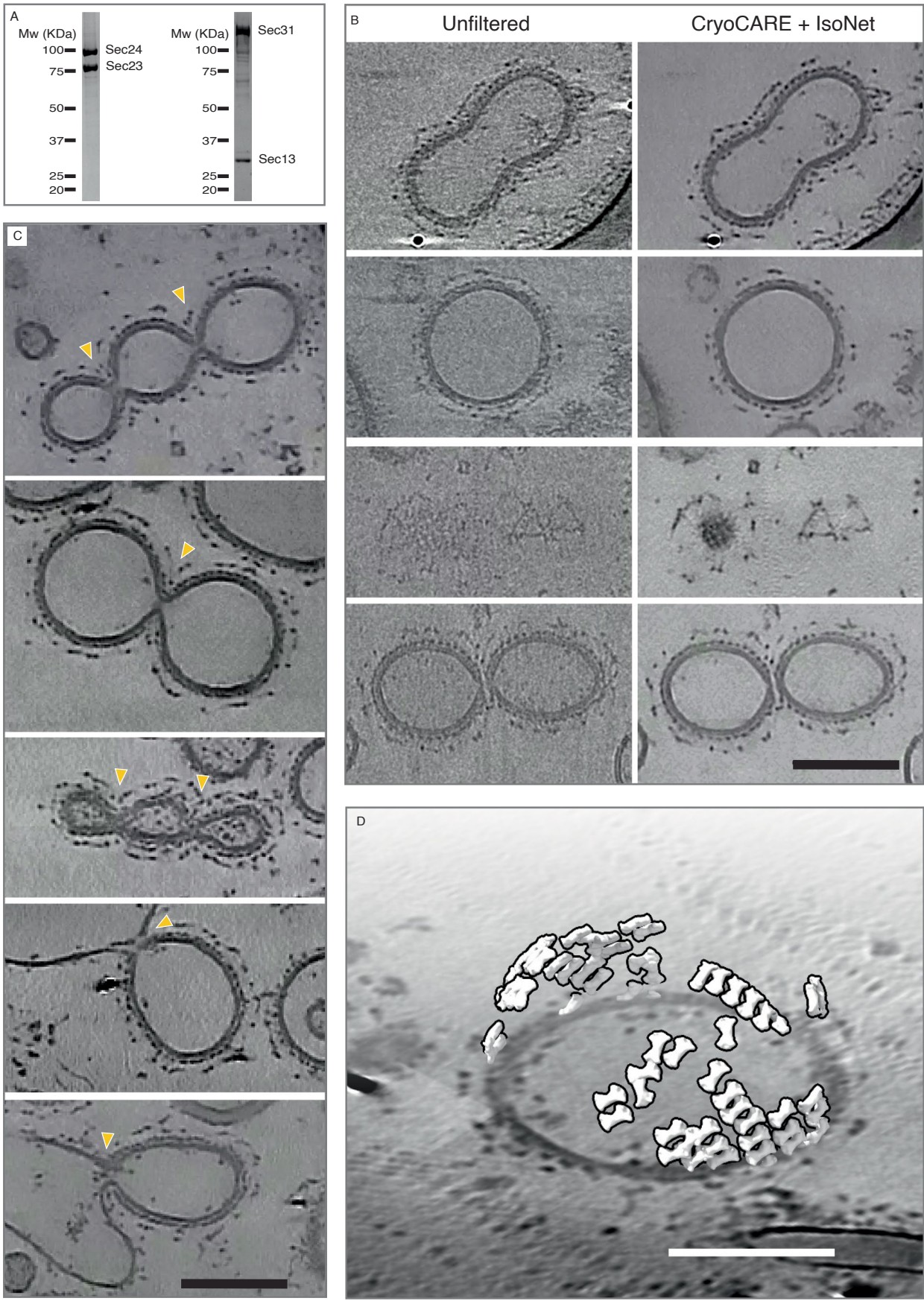

**Extended Data Fig. 1 | See next page for caption.**

**Extended Data Fig. 1 | Overview of the in vitro budding experiment. (A)** Polyacrylamide gel of purified Sec23-Sec24 and Sec13-Sec31 complexes from insect cell expression. **(B)** XY slices through reconstructed tomograms comparing unfiltered 8X binned tomograms (left panels) to the corresponding IsoNet-treated ones (right panels). **(C)** XY slices through reconstructed tomograms that show rare events with constricted but not detached vesicle necks (yellow arrowheads). **(D)** Placed inner coat subunits overlayed to an orthoslice of a representative tomogram. Scale bars 100 nm.

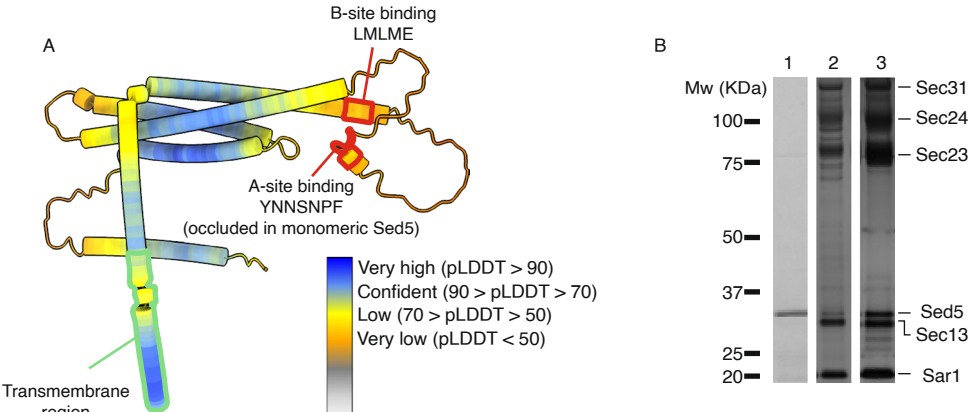

**Extended Data Fig. 2 | Characterisation of Sed5 cargo. (A)** Alpha-fold2 prediction of the full-length Sed5 structure (PDB ID: AF-Q01590-F1), coloured according to pLDDT value. The C-terminal transmembrane domain is highlighted in green, whilst the two Sec24-binding peptides are highlighted in red. These both fall within very low confidence regions, indicating disorder/flexibility. **(B)** Acrylamide gel showing purified Sed5 (lane 1), pelleted (lane 2) and floating (lane 3) fractions from a liposome flotation experiment.

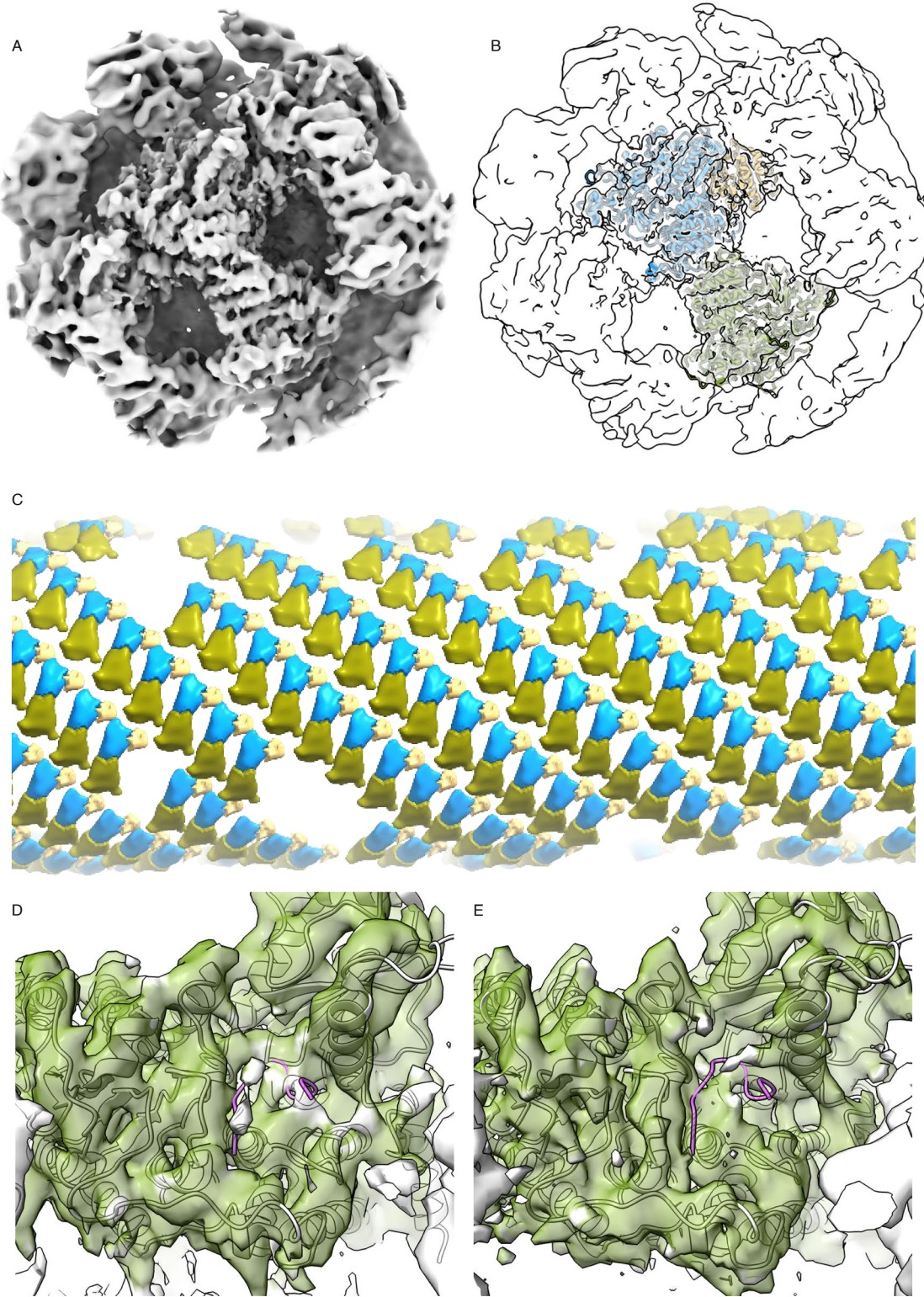

**Extended Data Fig. 3 | Subtomogram averaging of Sed5-bound inner coat.**
(**A**) Surface representation of the 4.1 Å STA map of the inner coat on Sed5-enriched GUV tubules (**B**) as in (**A**), with the Sec24-Sec24-Sar1 heterotrimer atomic model fitted. Sec23 in blue, Sec24 in green, Sar1 in yellow. (**C**) A 20 Å low-pass filtered map of the inner coat mapped back onto a representative section of a tomogram, showing the extensive lattice wrapping around a tubule. (**D**) and (**E**), As in Fig. 3E, F, but focussing on the A-site of Sec24, and showing the Sed5 YNNSNPF peptide bound as crystallised (PDB ID: 8PD0, in purple). While extra density in (**D**) (Sed5-bound) with respect to (**E**) cannot be excluded, it is difficult to unambiguously detect above the noise.

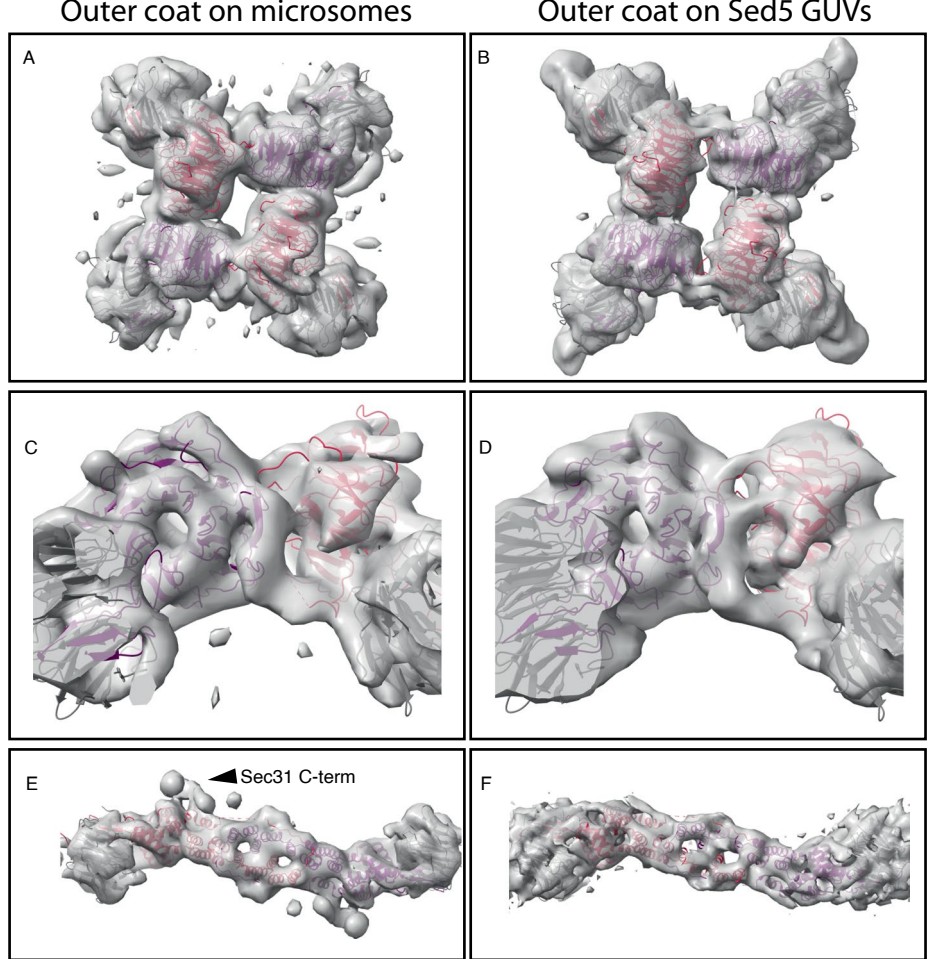

**Extended Data Fig. 4 | Comparison between the outer coat STA maps obtained from microsome and Sed5-GUV derived vesicles and tubes respectively. (A, B)** Overview of vertices, with four copies of the atomic model of the Sec13-Sec31 'vertex element' fitted (PDB 2PM9). Sec31 in red and purple, Sec13 in grey. **(C, D)** close up of the vertices from a side view. **(E, F)**, overview of rods, with the atomic model of the Sec13-Sec31 'edge' element fitted (PDB 2PM6). Colour code as in **(A, B)**.

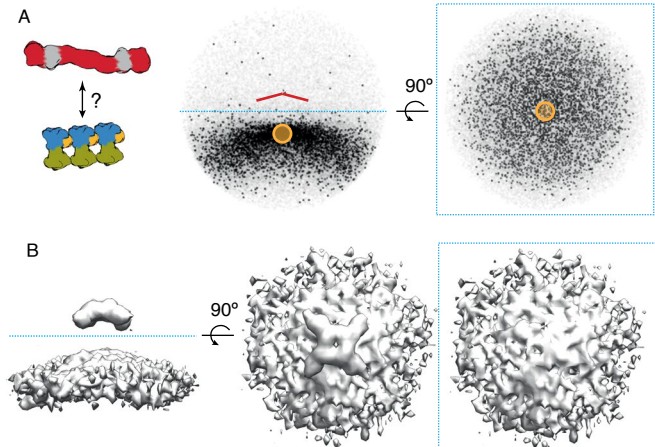

**Extended Data Fig. 5 | Relationship between inner and outer coat. A)** Plot of positions of aligned inner coat subunits with respect to each aligned vertex, viewed from the side (middle panel), and from the top (right panel). Each black 'dot' represents an inner coat neighbour. While inner coat subunits are at expected positions 'below' vertices, the absence of a pattern suggest random translation between the two lattices. **B)** Subtomogram average maps of vertices selected to have inner coat neighbours within the region shown in the yellow mask in (**A**), viewed from the side (left panel), from the top (middle panel), and from the top after removing the top half of the map (only showing density for the inner coat). The absence of any recognisable inner coat features indicated random rotation between the two lattices.

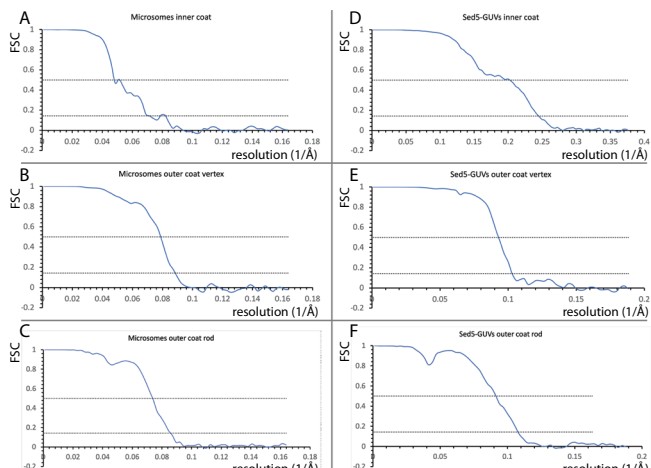

**Extended Data Fig. 6 | Fourier Shell Correlation plots for the maps discussed in this paper.** 0.5 and 0.143 thresholds are shown as dotted lines for reference. All refinements were carried out with two independent halves, and the resolutions reported are measured at the 0.143 threshold. (**A**) Inner coat on microsomes, (B) outer coat vertex on microsomes, (**C**) outer coat rod on microsomes, (**D**) inner coat on Sed5-GUVs, (**E**) outer coat vertex on Sed5-GUVs and (**F**) outer coat rod on Sed5 GUVs.

# Reporting Summary

## Statistics

For all statistical analyses, confirm that the following items are present in the figure legend, table legend, main text, or Methods section.

| n/a | Confirmed | |
|---|---|---|
| ☐ | ☒ | The exact sample size (*n*) for each experimental group/condition, given as a discrete number and unit of measurement |
| ☐ | ☒ | A statement on whether measurements were taken from distinct samples or whether the same sample was measured repeatedly |
| ☐ | ☒ | The statistical test(s) used AND whether they are one- or two-sided<br>*Only common tests should be described solely by name; describe more complex techniques in the Methods section.* |
| ☐ | ☒ | A description of all covariates tested |
| ☒ | ☐ | A description of any assumptions or corrections, such as tests of normality and adjustment for multiple comparisons |
| ☐ | ☒ | A full description of the statistical parameters including central tendency (e.g. means) or other basic estimates (e.g. regression coefficient) AND variation (e.g. standard deviation) or associated estimates of uncertainty (e.g. confidence intervals) |
| ☐ | ☒ | For null hypothesis testing, the test statistic (e.g. *F*, *t*, *r*) with confidence intervals, effect sizes, degrees of freedom and *P* value noted<br>*Give P values as exact values whenever suitable.* |
| ☒ | ☐ | For Bayesian analysis, information on the choice of priors and Markov chain Monte Carlo settings |
| ☒ | ☐ | For hierarchical and complex designs, identification of the appropriate level for tests and full reporting of outcomes |
| ☒ | ☐ | Estimates of effect sizes (e.g. Cohen's *d*, Pearson's *r*), indicating how they were calculated |

*Our web collection on statistics for biologists contains articles on many of the points above.*

## Software and code

Policy information about availability of computer code

| Data collection | SerialEM 4.1 beta, with PaceTOMO. |
|---|---|
| Data analysis | IMOD v4.11.15, Isonet v0.1, Dynamo v1.1.509, Relion v4.0, Relion v5.0 (4.1-alpha-1-commit-d2053c), https://github.com/EuanPyle/Membrane_Associated_Picking, MATLAB 2023a, Chimera, ChimeraX |

For manuscripts utilizing custom algorithms or software that are central to the research but not yet described in published literature, software must be made available to editors and reviewers. We strongly encourage code deposition in a community repository (e.g. GitHub). See the Nature Portfolio guidelines for submitting code & software for further information.

## Data

Policy information about availability of data

All manuscripts must include a data availability statement. This statement should provide the following information, where applicable:
- Accession codes, unique identifiers, or web links for publicly available datasets
- A description of any restrictions on data availability
- For clinical datasets or third party data, please ensure that the statement adheres to our policy

COPII Inner Coat on Vesicles: EMDB-19417

COPII Outer Coat (Rod) on Vesicles: EMD-19421

COPII Outer Coat (Vertex) on Vesicles: EMD-19418

COPII Inner Coat on Tubes with Sed5: EMDB-19410

COPII Outer Coat (Rod) on Tubes with Sed5: EMDB-19414

COPII Outer Coat (Vertex) on Tubes with Sed5: EMDB-19416
COPII Outer Coat (5-way vertex) on vesicles: EMDB-19879

# Research involving human participants, their data, or biological material

Policy information about studies with human participants or human data. See also policy information about sex, gender (identity/presentation), and sexual orientation and race, ethnicity and racism.

| | |
|---|---|
| Reporting on sex and gender | N/A |
| Reporting on race, ethnicity, or other socially relevant groupings | N/A |
| Population characteristics | N/A |
| Recruitment | N/A |
| Ethics oversight | N/A |

Note that full information on the approval of the study protocol must also be provided in the manuscript.

# Field-specific reporting

Please select the one below that is the best fit for your research. If you are not sure, read the appropriate sections before making your selection.

☒ Life sciences     ☐ Behavioural & social sciences     ☐ Ecological, evolutionary & environmental sciences

For a reference copy of the document with all sections, see nature.com/documents/nr-reporting-summary-flat.pdf

# Life sciences study design

All studies must disclose on these points even when the disclosure is negative.

| | |
|---|---|
| Sample size | Total numbers of contributing particles for subtomogram averaging are stated in Materials and methods. The sample size was determined by the availability of data collection time. |
| Data exclusions | Tilt series were discarded on occasions where accurate tomogram reconstruction could not be achieved. Tilt images were excluded if mistracked or if contaminating objects obscured the view. |
| Replication | For subtomogram averaging, all datasets were divided into two halves for independent processing . Map resolution was determined by half-map Fourier Shell Correlation consistent with gold standard methods. |
| Randomization | Particles falling into the two halves for independent processing were selected randomly by RELION5 intrinsic routines |
| Blinding | Not relevant, as data processing is done following automatic procedures |

# Reporting for specific materials, systems and methods

We require information from authors about some types of materials, experimental systems and methods used in many studies. Here, indicate whether each material, system or method listed is relevant to your study. If you are not sure if a list item applies to your research, read the appropriate section before selecting a response.

## Materials & experimental systems

| n/a | Involved in the study |
|---|---|
| ☒ | ☐ Antibodies |
| ☐ | ☒ Eukaryotic cell lines |
| ☒ | ☐ Palaeontology and archaeology |
| ☒ | ☐ Animals and other organisms |
| ☒ | ☐ Clinical data |
| ☒ | ☐ Dual use research of concern |
| ☒ | ☐ Plants |

## Methods

| n/a | Involved in the study |
|---|---|
| ☒ | ☐ ChIP-seq |
| ☒ | ☐ Flow cytometry |
| ☒ | ☐ MRI-based neuroimaging |

## Eukaryotic cell lines

Policy information about cell lines and Sex and Gender in Research

| | |
|---|---|
| Cell line source(s) | Sf9 insect cells for expression are from invitrogen |
| Authentication | No authentication done |
| Mycoplasma contamination | Not assessed |
| Commonly misidentified lines (See ICLAC register) | No commonly misidentified cell lines were used |

## Plants

| | |
|---|---|
| Seed stocks | N/A |
| Novel plant genotypes | N/A |
| Authentication | N/A |

