## [Peer Review File · Nature Structural & Molecular Biology]

Cryo-electron tomography reveals how COPII assembles on cargo-containing membranes

Corresponding Author: Dr Giulia Zanetti

A version of this paper was originally rejected for publication by Nature Structural & Molecular Biology, however that decision was reconsidered after appeal by the authors.

Version 0:

Decision Letter:

23rd Feb 2024

Dear Dr. Zanetti,

Thank you for submitting your manuscript "Cryo-electron tomography reveals how COPII assembles on cargo-containing membranes". I apologize for the delay in processing your manuscript, which resulted from difficulties in obtaining referees' reports. Nevertheless, we now have comments from the 3 reviewers who have evaluated your manuscript are below. Unfortunately, after carefully considering their comments, we cannot offer to publish your manuscript in Nature Structural & Molecular Biology.

You will see that while the referees find the work of potentially interesting, they raise concerns about the depth of structural characterisation of the observed COPII coat architectures.

However, if further experimentation, analysis, and revisions allow you to address the referees concerns in full, we would be prepared to consider an appeal of our decision, on the condition that no related work is published in the interim or has been accepted in our journal. Please contact me to discuss an appeal and potential revision. We would be happy to schedule a call to discuss the options. Please note that, until we have the opportunity to read the revised manuscript in its entirety, we cannot promise that it will be sent back for peer review.

The required new experiments and data include, but are not limited to, further structural analysis, as requested by reviewer #1. Moreover, we would ask that you address the comments of reviewer #2 about the potential for non-physiological scission, as well as expand the analysis to provide more insights into the outer and inner coat interactions. We agree with reviewer #2 that while testing mutants would strengthen the manuscript, we will not consider it necessary. We also ask that you address all technical concerns brought up by referee #3.

I am sorry we could not be more positive on this occasion. I hope that you find the referees' comments useful in deciding how best to proceed.

Sincerely,
Kat

Katarzyna Ciazynska, PhD
(she/her)
Associate Editor
Nature Structural & Molecular Biology
<https://orcid.org/0000-0002-9899-2428>

Referee expertise:

Referee #1: COPII

Referee #2: COPII

Referee #3: cryo-ET and subtomogram averaging

Reviewers' Comments:

Reviewer #1:

Remarks to the Author:

This paper describes new advances from this group using their excellent experimental system of COPII-coated transport carriers analyzed by cryo-electron tomography.

The new aspects and observations are:

- (1) The use of a natural membrane source, yeast microsomes, for COPII carrier formation, contrasting with their prior work using synthetic liposomes.
- (2) The consequent synthesis of dominantly spherical vesicles rather than tubular containers.
- (3) The arrangement of inner-coat complexes on these spherical vesicles.
- (4) An assessment of cargo binding: the detection of cryo-tomography density features corresponding to cargo.
- (5) The observation of a wide range of cage/coat designs, including novel 5-fold cage vertices.

The main findings of interest in the study are: First, the arrangement of inner-coat protein on spherical vesicles. These are shown to organize as localized patches; this is a nice advance and rationalization of our understanding.

Second, the finding of a diversity of coat/cage structures, including the description of elbow interactions and the 5-fold symmetry centers of the cage vertex. I would like to see a much more thoroughgoing analysis of the varieties of COPII structures. The gold standard study, in this context, is the 2007 J Mol Biol study by Yifan Cheng, Steve Harrison and Tom Walz, in which they used cryo-electron tomography to analyze clathrin-coated vesicles purified from brain. A range of cage designs were defined. The distinctive element in that study is that a series of designs are analyzed individually in terms of their size, geometry/symmetry and architecture. As a result, the paper yields deep insight into clathrin/adaptin architecture.

This level of detail is absent from the present manuscript, and my enthusiasm for publication in NSMB is lowered as a result.

Minor points:

Cargo:

The analysis of cargo-binding sites draws on prior observations, as the authors acknowledge. The experiments (including Sed5 addition) are nicely done. While it is interesting to see density appear in the cryo-tom maps at known cargo-binding sites on COPII, this is just a demonstration of prior knowledge. These sites were identified originally by high-resolution structural analysis and functional analysis in yeast transport experiments. Consequently, the observation of low-resolution density features at the B-site of Sec24 is not novel. The absence of density for Sed5 at the A-site is referred to as "interesting", but I really think this should be considered "concerning". This is a bona fide site and interaction validated by budding experiments in yeast. The authors should offer more analysis and information on this contradictory result.

Reviewer #2:

Remarks to the Author:

The work by Pyle and Zanetti is a beautiful follow up to prior work in the Zanetti lab describing the structure of the COPII coat assembled on membranes. Here, they use ER microsomes, as opposed to synthetic GUVs, to demonstrate that yeast COPII largely generates spherical vesicles, as opposed to tubules described in their earlier work. The distribution of the inner COPII coat on vesicles is interesting, suggesting that an intact inner layer is unnecessary for vesicle formation. Studies with the Sed5 cargo are compelling, and the findings shown are logical (although not particularly novel). New insights into the structure of the outer coat are also highlighted. However, the authors should address a few shortcomings prior to publication of this work:

1. The authors claim that vesicle scission mediated by COPII is independent of GTP hydrolysis. However, the incubation period of purified COPII with microsomes and GMP-PNP is extensive (i.e., 30 min), much longer than the time required for budding under physiological conditions. It would be best to examine budding following shorter periods of incubation (i.e., 1 min or 5 min) to see whether free vesicles are formed. If not, it is likely that the prolonged incubation with active Sar1 results in a non-physiological scission reaction.
2. The authors should extend their analysis to examine the interaction between the inner and outer layers of the COPII coat. It is unclear why this is not explored in the current manuscript. In particular, are the patches of inner coat observed typically associated with the outer coat? In what manner? Do the inner coat patches form in the absence of the outer coat? Does the heterogeneity of the outer coat correlate with the presence/absence of underlying inner coat?
3. A mutant form of Sed5 (harboring mutations in its A- and B-sites) should ideally be tested to determine its distribution relative to the inner coat, although this is not absolutely necessary.

Reviewer #3:

Remarks to the Author:

Summary: This manuscript examines the structure and organization of the COPII lattice on cell-derived membranes, building upon previous high-resolution tomography results on COPII structure by the same lab on in vitro reconstituted lattices on GUVs. The authors identify a fragmented organization in spherical ER-derived vesicles, which contrasts with the more idealized lattices previously solved on GUVs. The authors use both systems in this manuscript, employing the native-like membranes for insight into COPII organization and then moving to the GUV ordered assembly for higher resolution structures. Biological insights presented in this work include arguments that the inner coat interactions with native lipids and cargo influence curvature, and that the patchy native lattice allows room for cargo binding and presumably a greater diversity of cargoes than the ordered helical assembly on GUVs would.

Overall evaluation: The study is well designed in terms of rationale and methods, making good use of a combination of idealized and more native-like systems to balance biological insight with resolution. In cases where higher resolution is not possible, such as cargo identification, the authors use reasonable controls (such as the use of subtraction maps against GUVs devoid of cargo). Some key points need to be clarified in the methods, and in general the paper would benefit from greater discussion of previous work and alternative interpretations. I recommend publication following revisions.

Major Points to Address:

1. The microsome system needs to be better described, in particular whether ER-derived microsomes have intrinsic vesicle diameters or curvature patches in the absence of COPII, and whether major protein components present have been identified. Hopefully this was already done in previous publications and just needs to be clarified in the text. If not, it would aid the paper to include mass spectrometry identification of major protein components in the microsomes, for example to confirm that cargo is likely the density described in Figure 3 rather than a cofactor.
2. The new lattice arrangement is a very interesting result, but still leaves the question of whether the native lipids are now resulting in a native-like assembly. I do NOT propose that the authors perform in situ tomography themselves (in my opinion that is unreasonable), but are tomograms available in EMPIAR that they can search and confirm general similarity in COPII patchy coatings of vesicles, or other published evidence they can use to bolster claims of physiological relevance? Perhaps the chlamydomonas dataset from Engel et al, EMPIAR 11756?
3. The current methods sound like the authors are using denoised tomograms for subtomogram averaging - hopefully there is a missing step in the method description? I would like the authors to confirm that denoised tomograms are used to identify particle positions, but then the particles are extracted from non-denoised tomograms prior to alignment and averaging.
4. This is related to point #3: the authors are deriving meaning from the absence of particles, which requires high confidence in the accuracy of particle identification. IsoNet is a powerful denoiser, but it achieves this goal in part through a de facto averaging along surfaces, which can lead to induced density or lack thereof for membrane-bound particles. As a control, I would like to see in the supplement orthoslices of denoised and non-denoised tomograms for the microsome COPII assemblies, with particle picking comparisons of COPII patches showing that picking is facilitated by denoising but not systematically affected (for example, empty patches of COPII because the denoiser erased them, particularly in areas most heavily affected by the missing wedge artefact). I would also like to see estimates for the accuracy of the particle identification to confirm that expected error would not affect the conclusions - this may be in the rebuttal only if the authors cannot make a robust analysis for publication.

List of additional points to consider by line:

- Line 99: Blotting generates shearing forces that are probably strong enough to rip a narrow neck. This doesn't mean this interpretation is incorrect, but please reword to include the possibility of this mechanical perturbation's influence on the result.
- Figure 2: please put one of these models on top of an orthoslice, which can show the reader what the blank spaces look like
- Figure 2: is there any data elsewhere, such as mass spectrometry-based stoichiometries, to confirm the finding that the vesicles are only half-occupied by COPII? This would be very helpful to cite in the discussion.
- In general the argument regarding the Sed5 and Figure 3 gets rather murky — cargo binding doesn't disrupt the lattice, but the lattice isn't a strong feature in physiologically-relevant lipids based on the ultrastructure? This section could use some attention to tighten the argument.
- Section beginning at Line 136: please include more discussion regarding the assumed cargo density (+/- Sed5), for example whether there could be subunit flexibility due to changes in lipids, native cofactors, etc. I agree that the authors' interpretation is the most likely, but other interpretations are possible and should be addressed. Have they been ruled out, and if so why?
- Line 248 and Figure 4E: have these hinge regions been previously identified, or predicted based on sequence/AlphaFold? Please mention any previous findings or predictions.
- Lines 253, 262: I think "circles" would be more clear than "lozenges" for the majority of readers
- Line 270: "which largely comprise of the ER" - awkward wording. Which are largely comprised, perhaps?
- Line 332 (Sec 23/24, Sec 13/31, Sar1 purification): the cited paper has a mixture of cleaved and uncleaved His tags. Please clarify whether His tags were present here.
- Section beginning at Line 420 (tomography data collection): please state the number of tomograms collected and the number included in the final particle datasets.
- Please state the number of particles used to generate the reference, and confirm that they are not included in the subsequent subtomogram averaging (to maintain independence in halfsets).
- Please add FSC traces to the supplement for all stated resolutions.
- Line 499 and elsewhere: please include the resolution of the reference used in template matching, and confirm that final

resolution of the subtomogram average surpassed the resolution of the reference (or other approaches to guard against Einstein from Noise artefacts).

•Line 584: I'm assuming that the inner coat must be present for the outer coat to bind, and this is why the positions were not searched independently? Please state this clearly in the methods.

** For Springer Nature Limited general information and news for authors, see <http://npg.nature.com/authors>.

Version 1:

Decision Letter:

1st May 2024

Dear Dr. Zanetti,

Thank you again for submitting your manuscript "Cryo-electron tomography reveals how COPII assembles on cargo-containing membranes". I apologize for the delay in responding, which resulted from the difficulty in obtaining suitable referee reports. Nevertheless, we now have comments (below) from the 3 reviewers who evaluated your paper. In light of those reports, we remain interested in your study and would like to see your response to the comments of the referees, in the form of a revised manuscript.

You will see that while the referees appreciate the efforts put into the revision, they have some remaining concerns. Specifically, we note that reviewer #3 has remaining technical concerns regarding the approaches taken in the data processing, which need to be addressed before we can further consider the manuscript.

Please be sure to address/respond to all concerns of the referees in full in a point-by-point response and highlight all changes in the revised manuscript text file. If you have comments that are intended for editors only, please include those in a separate cover letter.

We expect to see your revised manuscript within 6 weeks. If you cannot send it within this time, please contact us to discuss an extension; we would still consider your revision, provided that no similar work has been accepted for publication at NSMB or published elsewhere.

Please let us know if you would like to discuss this decision and we can schedule a phone call.

Reporting Summary:

Please note that all key data shown in the main figures as cropped gels or blots should be presented in uncropped form, with molecular weight markers. These data can be aggregated into a single supplementary figure item. While these data can be displayed in a relatively informal style, they must refer back to the relevant figures. These data should be submitted with the final revision, as source data, prior to acceptance, but you may want to start putting it together at this point.

Data availability: this journal strongly supports public availability of data. All data used in accepted papers should be available via a public data repository, or alternatively, as Supplementary Information. If data can only be shared on request, please explain why in your Data Availability Statement, and also in the correspondence with your editor. Please note that for some data types, deposition in a public repository is mandatory - more information on our data deposition policies and available repositories can be found below:

<https://www.nature.com/nature-research/editorial-policies/reporting-standards#availability-of-data>

Link Redacted

Sincerely,

Katarzyna Ciazynska, PhD
(she/her)
Associate Editor
Nature Structural & Molecular Biology
<https://orcid.org/0000-0002-9899-2428>

Reviewers' Comments:

Reviewer #1:

Remarks to the Author:

The authors should be commended for their thorough and scholarly effort to improve the manuscript. I reiterate from my original review that the observation of small patches of Sec23/24 with the rhomboidal lattice form is a very nice observation. And the authors conclusion that this lattice is compatible with small but not large cargo, as an explanation for the preponderance of vesicles in their microsome-derived prep, makes good sense. This is a nice advance in

understanding.

At this point, I offer a (long) list of comments for the authors to consider and I leave other decisions to the editorial staff. I am not insisting on these changes, I just propose points on which the clarity of the science or the grammar could be enhanced, in my view.

Abstract:

Line 16, change "via" to "through".

Line 18, change "wide ranges" to "a wide range". (When preceded by "a", "range" becomes plural).

Line 19. The statement that "the cargo-dependent regulation of [COPII coat] assembly remain[s] poorly understood" belies some significant old observations. E.g. Yeung, Barlowe and Schekman showed (JBC, 1995) that ER membranes thoroughly depleted of cargo proteins are competent for packaging and COPII budding of machinery proteins (SNAREs). So, perhaps there is no cargo-dependent regulation of COPII assembly. Work on Tango1, by contrast, indicates possibly a major role in COPII assembly; but Tango1 is not a cargo molecule. I recommend that the authors tighten up their wording. Perhaps they mean to refer to how cargo may influence the form of the vesicle.

Line 27. I make no objection to – in fact I welcome – the author's hypothesizing on the underlying mechanism for curvature generation. However, "challenge[s] our current understanding" strikes a false note (the word grandiloquent comes to mind). If the majority of COPII researchers were asked, I suspect there would be minimal consensus on this issue, hence there is little to be challenged. The scientific problem is that since all the proteins (Sec23, 24, 13, 31, Sar1) have so thoroughly co-evolved, attempting to delineate and ascribe curvature generation to any one component is most likely an insoluble problem.

Results:

Line 153, The authors show density for the A- and B-sites, but not the C- and D-sites. I have no particular criticism here, as I assume binding is substoichiometric to the latter sites (the C-site may well be uniquely the binding site for Sec22), whereas a broad range of cargoes likely binds to the A- and B-sites. But perhaps the authors should make mention of this issue (beyond the statement in line 174).

Line 157, I am at a loss as to the purpose of the mass spectrometry experiment. The microsomal prep contains ER proteins, that is its nature, and this will include the ER export receptors listed in this paragraph, since these proteins recycle and will be at appreciable steady-state levels in the membranes. I'm sorry to be harsh, but measuring by mass spec the content of starting membranes makes no useful contribution to this study. Even measurement of the derived vesicle membrane contents would be no more than a demonstration experiment. Finally, without quantitation, there is no way to decide whether the post-ER residents detected in the microsomal prep are contaminants. Indeed, in the rebuttal, the authors state that "mass spectrometry detected almost 8000 proteins". Perhaps I am missing something, but I didn't think there were that many protein-coding genes in yeast.

Line 180-181, The availability of Sed5 motifs for binding COPII was demonstrated in biochemical experiments not "X-ray studies".

Line 183, In addition to deploying alphafold, perhaps the authors should reference two formative studies here (from Fred Hughson and Josep Rizo/Tom Sudhof) which established the conformational switching of Syntaxin proteins and thereby predicted Sed5 behavior (Munson et al, & Hughson, 2000, Nat. Struct. Biol.; Dulubova et al. & Rizo, 1999, Embo J).

Line 200, Regarding Sed5, I am pleased to see a more complete description conveyed in the manuscript. I didn't see this in the previous version – perhaps I overlooked it. However, although the authors introduce this section with a mention of conformational switching (Line 181), they have not drawn any conclusion regarding the absence of density at the A-site in their study. Does their data conform to the notion of COPII having specificity for distinct states of the SNARE, or do the authors think not? In the rebuttal, it is stated: "Previous studies have shown that this peptide is occluded in the monomeric form of Sed5, and it only becomes exposed when Sed5 is in the Golgi SNARE complex with Bet1, Bos1 and Sec22". What a clear and coherent statement; but do the authors not believe this sufficiently to state it in the manuscript proper? Some comment on this issue seems to me to be required. (As an aside, I believe the early study showed availability of the Sed5 YNNSNPF motif in the t-SNARE as well as the v-/t-SNARE complex.

Figure 4 is very nice. The clear statement of lack of symmetry of individual cages is well made.

Line 274, I would have to think this through further, but it occurs that there is a question as to what is happening structurally at the fivefold centers. I understand that the resolution does not allow for the details to emerge. However, the question is whether this is (a) quasi-equivalence whereby, in the classical meaning of the phenomenon, there is structural polymorphism of the Sec31 beta-propeller such that it can form a discrete 5-fold vertex arrangement; or (b) the Sec31 ends of the five rods are being "pushed" or held together by the sufficient driving stability of the remainder of the polyhedron, such that there is in fact not a "special" 5-fold interface arrangement. If the authors think there is some sense to what I have written (and I'm not certain there is) they may wish to make mention of this ambiguity.

Line 322, recommend to remove the words "without breaking".

Line 358, The authors have been lured by referee into questioning whether their observations might be confounded by use of the non-hydrolyzable GTP analogue. This is unfortunate, and commendably the authors have stuck to their view. This

issue was dealt with directly in a study from the lab of Felix Wieland. In fact, the sole purpose of that study was to address whether vesicle release requires GTP hydrolysis. It does not. The reference is Adolf, et al. & Wieland, 2013, Traffic "Scission of COPI and COPII vesicles is independent of GTP hydrolysis".

Reviewer #2:

Remarks to the Author:

The authors have nicely addressed concerns raised during initial review, and the manuscript now appears ready for publication.

Reviewer #3:

Remarks to the Author:

Large overview of the manuscript revisions: in general the rigor of the response to reviewer suggestions is underwhelming and has dampened my enthusiasm for publication in a higher-tier journal like NSMB.

Points still at issue following revision:

Major points 1 and 2 have not been robustly addressed. Uncoated microsome diameters etc. were not measured, and the authors did not propose a different way of showing physiological relevance in the absence of EMPIAR tomography data. The published reference image they included, which was identified as COPII in the eLife paper, appears more ordered than the authors' results.

On the data processing side:

-Some aspects of the processing approach are controversial. Specifically, the use of particles in both the initial reference and the dataset and the use of higher-resolution references to aid particle picking. The authors mention the RELION workflow as requiring the inclusion of particles from the reference in the dataset, perhaps they do not realize there are other options possible within that software environment? Regardless, these controversial approaches can be validated by the final refinement of a high-resolution structure with features far exceeding the initial reference, and clean FSCs. In this case, the tests do not pass: use of a 25Å reference results in a 14Å map and the FSCs have issues. One of the new FSCs does not cross the x-axis; another has a shoulder that is likely either the mixture of a subset of noise-aligned "particles" with real data or overfitting of flexible protein. These may indicate problems with the data processing, discussed above.

-IsoNet and particle picking validation was not well addressed in the rebuttal and revision. Confirming that manual picks from controls are also found by IsoNet-facilitated picking is a poor validation because particles easily seen by the naked eye would be expected to come through any processing approach. Independence of orientation and regularity are certainly good, but not enough to validate a neural network that can impose regularity as part of its algorithm. This would not be a problem for a structure solve, but it is a problem for deriving meaning from the positions and orientations of particles. Stronger discussion of strengths and weaknesses, and tempering of conclusions based on them, is warranted.

-The authors are still not making the distinction between the number of tomograms collected/reconstructed and number of tomograms contributing to the final analysis.

Regarding the conclusions drawn from the structures, several points have not been addressed adequately:

-Shearing forces as a result of blotting are still not mentioned in the manuscript as a possible contributor to scission.

-Hinges that were not previously indicated in other publications should not be newly identified here from low-resolution structural data.

-Sed5 identity caveats are still missing from the manuscript

-The authors acknowledge that the inner coat is not being identified well in the particle picking. In this case, how can the authors be sure that the minority of inner coat particles they are identifying aren't inherently different, thus making them amenable to picking? And how can they have confidence that the inner coat isn't necessary for COP II outer coat when they are having trouble identifying the inner coat in the first place?

-The statements the authors are making about the inner coat's positioning are too strong considering they are missing the majority of the data.

Version 2:

Decision Letter:

19th Jun 2024

Dear Dr. Zanetti,

Thank you again for submitting your manuscript "Cryo-electron tomography reveals how COPII assembles on cargo-containing membranes". I apologize for the delay in responding, which resulted from the difficulty in obtaining suitable referee reports. As you know, we have sought arbitration to settle the disagreement with reviewer #3 regarding the processing approaches undertaken. The comments of the arbitrating referee #4, who evaluated the paper are below. In light of the report, we remain interested in your study. However, in line with reviewer comments we ask for an effort on your part in revising the manuscript to address the issues brought up with the FSC plots, specifically in supplementary figure 6 C and F.

Please be sure to address/respond to all concerns of the referees in full in a point-by-point response and highlight all changes in the revised manuscript text file. If you have comments that are intended for editors only, please include those in a separate cover letter.

We appreciate that you have already been through an extensive review and revision process. We are committed to providing a fair and constructive peer-review. Do not hesitate to contact us if there are specific requests from the reviewers that you believe are technically impossible or unlikely to yield a meaningful outcome. Please let me know if you would like to schedule a call to discuss any of the points, and our editorial expectations of the revision.

We expect to see your revised manuscript within 6 weeks. If you cannot send it within this time, please contact us to discuss an extension; we would still consider your revision, provided that no similar work has been accepted for publication at NSMB or published elsewhere.

Reporting Summary:

Please note that all key data shown in the main figures as cropped gels or blots should be presented in uncropped form, with molecular weight markers. These data can be aggregated into a single supplementary figure item. While these data can be displayed in a relatively informal style, they must refer back to the relevant figures. These data should be submitted with the final revision, as source data, prior to acceptance, but you may want to start putting it together at this point.

Data availability: this journal strongly supports public availability of data. All data used in accepted papers should be available via a public data repository, or alternatively, as Supplementary Information. If data can only be shared on request, please explain why in your Data Availability Statement, and also in the correspondence with your editor. Please note that for some data types, deposition in a public repository is mandatory - more information on our data deposition policies and available repositories can be found below:

<https://www.nature.com/nature-research/editorial-policies/reporting-standards#availability-of-data>

Link Redacted

Sincerely,

Katarzyna Ciazynska, PhD
(she/her)
Associate Editor
Nature Structural & Molecular Biology
<https://orcid.org/0000-0002-9899-2428>

Referee expertise:

Referee #4: cryo-ET and subtomogram averaging

Reviewers' Comments:

Reviewer #4:

Remarks to the Author:

In this manuscript, Pyle and colleagues determine the structure of COPII assemblies generated using protein from *S. cerevisiae* and native membranes. Previous structures of COPII-coated membranes determined by this group showed that COPII forms a pseudo-helical lattice on tubular vesicles. However, these prior studies reconstituted COPII assemblies using Giant Unilamellar Vesicles (GUVs) generated from purified lipids. These tubular vesicles appear distinct from the spherical COPII vesicles that have been seen in cellular tomograms. To address this potential problem, the authors here reconstitute COPII assemblies using native membranes purified from yeast ER membranes (i.e. microsomes), and find that these assemblies take on a spherical morphology that more closely resembles those seen in situ, rather than those assemblies from GUVs. The authors determine the structures of the inner COPII assembly, and the outer coat centered on either vertices or rods; each of these structures were determined for assemblies reconstituted from microsomes or Sed5-enriched GUVs.

As this manuscript has already gone through at least one round of revision, I will mainly focus my review on the technical aspects of the subtomogram averaging (STA) used to determine the structures presented.

With regards to processing workflow and dataset size, (i.e. number of tomograms collected/used and remaining subtomograms) I think at least a supplementary table would make things easier to follow; a processing workflow diagram showing the steps of the workflow and packages used would be better.

In a previous round of review, reviewer 3 referred to some aspects of the processing approach here as "controversial". Specifically, this statement seems to refer to two things: using particles from the initial reference in the full dataset; and using a "higher-resolution" reference for particle picking.

On the first point, I don't think I've ever seen a STA manuscript that uses an independent subset of the data purely for picking. While there may be some issues with the so-called "attraction problem" where a particle correlates against itself in the averaged reference, this likely has a minor effect in the dataset sizes used here; the use of a maximum likelihood algorithm (i.e. RELION) is more likely to bias the results (see: <https://doi.org/10.1016/j.jsb.2018.08.002>). Viewing this from first-principles, this point is largely moot as every iterative alignment algorithm at the core of STA or single particle analysis (SPA) uses references generated from the particles.

On the second point, I think "higher-resolution" is a bit poorly defined to be used in a cogent argument. Any useful template for template matching must be "higher-resolution" than the raw tomographic data and isotropically-resolved; STA is

effectively a pre-requisite. A better way to think about this is what resolution limit is reasonable for template matching without imposing too much reference bias. As I noted in the point above, this is again a fundamental problem of iterative alignment. Here, there are no hard-and-fast rules, but best practices include so-called "gold standard" processing using half maps and low-pass filtering references. "Gold standard" is where the dataset is split in half, and each is refined independently using independent references. However, this splitting is typically performed later in the processing pipeline after particle picking is performed (e.g. 3D auto-refine in RELION); the template matching approach used here follows this. As for low-pass filtering, the ideal approach is to use the lowest resolution information possible. This is rarely known a priori, so another approach is to determine if the resolution of the map is sufficiently beyond the low pass filter limit used for alignment. In the case cited here of a 25Å map yielding a 14Å structure, we can estimate the increase in information content in Fourier space. This shows the information is at 0.040 1/Å and 0.071 1/Å respectively, which is an approximate 1.78 fold increase in information in one dimension and a 5.6 fold increase in 3D information. Looking at the FSC plots in supplementary figure 6, the FSCs of all structures is still at ~1 at the low pass filter cutoff at 0.04 1/Å; this seems completely reasonable since high-resolution refinement will typically push this much further.

I do have some concerns with the FSC plots shown in supplementary figure 6 C and F. For C, the FSC seems to only pass 0 for 1 Fourier shell while in F it never passes 0. For a well-determined structure the FSC should reach 0 and oscillate around 0, as true noise shouldn't correlate; correlation past the estimated resolution can be a sign of overfitting. This ideal behavior is seen in all the other FSCs presented, which may suggest some issue specifically in the outer coat rod processing. I don't agree with Reviewer 3's notion that a shoulder here is related to noise alignment, as I don't see any theoretical basis to this argument. What seems likely to me for C and F are that there are duplicate particles between the halfsets. While the authors mention removing duplicates, I can imagine RELION pushing particles towards neighboring "shiny" particles, effectively resulting in duplication. In any case, this issue needs to be resolved; suggesting that the FSC staying below 0.143 is sufficient is not a reasonable argument.

For the point on validating manual picking, I don't think there's any good approach for this as real data does not have a "ground truth". Ultimately, 3D classification is the only practical approach for removing false positives, with the caveat that it is also not perfect. I think reviewer 3's issues with neural network hallucinations are a non-issue, as IsoNet data was not used for image processing.

In regards to the hinges described in Fig 5, I am a bit unclear about some of the data being shown. For Fig 5A, are the density maps simply illustrations of how rods can bend around vertices, or are they actually averaged from the data? As for the hinges described in C and D, this seems like a fair comparison, but more information is needed. Specifically, how many particles in each class, resolution of each class, and whether the overlaid maps shown are filtered to the same resolution.

Overall, I think the processing pipelines used in this manuscript are pretty standard for the field. While I have described several points that need clarification, I don't see these issues substantially affecting the findings in this manuscript.

Version 3:

Decision Letter:

Our ref: NSMB-A48726C

5th Jul 2024

Dear Dr. Zanetti,

Thank you for submitting your revised manuscript "Cryo-electron tomography reveals how COPII assembles on cargo-containing membranes" (NSMB-A48726C). It has now been seen by the original referees and their comments are below. The reviewers find that the paper has improved in revision, and therefore we'll be happy in principle to publish it in Nature Structural & Molecular Biology, pending minor revisions to satisfy the referees' final requests and to comply with our editorial and formatting guidelines.

We are now performing detailed checks on your paper and will send you a checklist detailing our editorial and formatting requirements in about a 2-3 weeks. Please do not upload the final materials and make any revisions until you receive this additional information from us.

To facilitate our work at this stage, it is important that we have a copy of the main text as a word file. If you could please send along a word version of this file as soon as possible, we would greatly appreciate it; please make sure to copy the NSMB account (cc'ed above).

Sincerely,
Kat

Katarzyna Ciazynska, PhD

(she/her)
Associate Editor
Nature Structural & Molecular Biology
<https://orcid.org/0000-0002-9899-2428>

Reviewer #4 (Remarks to the Author):

In this revision, Pyle and colleagues have addressed the comments in my prior review, which focused on some clarifications in the methods and text, and more importantly addressing some concerns I had with some of their results.

For the comments on clarification, this included the my request for a table outlining their data processing and additional text clarification on panels in Figure 5. These issues have been satisfactorily addressed.

As for results, I was concerned about the FSC plots shown in supplementary figure 6, panels C and F. Specifically, they did not drop and stay around zero, suggesting some spurious but systematic problem. While I suggested that there may be an issue with identical particles in the half-sets, the authors noted that the systematic error was potentially related to the shape of the mask they used, and they recalculated the FSCs using a spherical mask. This seems to have addressed the issue, as the FSCs now look as expected.

I have no further concerns or comments.

Version 4:

Decision Letter:

1st Oct 2024

Dear Dr. Zanetti,

We are now happy to accept your revised paper "Cryo-electron tomography reveals how COPII assembles on cargo-containing membranes" for publication as an Article in Nature Structural & Molecular Biology.

Your paper will be published online soon after we receive proof corrections and will appear in print in the next available issue. You can find out your date of online publication by contacting the production team shortly after sending your proof corrections.

You may wish to make your media relations office aware of your accepted publication, in case they consider it appropriate to

organize some internal or external publicity. Once your paper has been scheduled you will receive an email confirming the publication details. This is normally 3-4 working days in advance of publication. If you need additional notice of the date and time of publication, please let the production team know when you receive the proof of your article to ensure there is sufficient time to coordinate. Further information on our embargo policies can be found here:
<https://www.nature.com/authors/policies/embargo.html>

Please note that *Nature Structural & Molecular Biology* is a Transformative Journal (TJ). Authors may publish their research with us through the traditional subscription access route or make their paper immediately open access through payment of an article-processing charge (APC). Authors will not be required to make a final decision about access to their article until it has been accepted. [Find out more about Transformative Journals](https://www.springernature.com/gp/open-research/transformative-journals)

Sincerely,

Katarzyna Ciazynska, PhD
(she/her)
Senior Editor
Nature Structural & Molecular Biology
<https://orcid.org/0000-0002-9899-2428>

We would like to thank all reviewers for very insightful comments, which we believe helped us improve our manuscript.

In particular, reviewer 1 and 2 pointed to lack of analysis of outer coat cages and of the relationship between the inner and outer coat. We believe the addition of these analyses in the revised manuscript has helped significantly broadening its scope. Addition of proteomics data as requested by reviewer 2 also helps strengthen our claims of cargo incorporation. Finally, we would like to thank reviewer 3 for their thorough feedback, we believe the manuscript has significantly improved in clarity as a consequence.

Reviewer #1:

Remarks to the Author:

This paper describes new advances from this group using their excellent experimental system of COPII-coated transport carriers analyzed by cryo-electron tomography.

The new aspects and observations are:

- (1) The use of a natural membrane source, yeast microsomes, for COPII carrier formation, contrasting with their prior work using synthetic liposomes.
- (2) The consequent synthesis of dominantly spherical vesicles rather than tubular containers.
- (3) The arrangement of inner-coat complexes on these spherical vesicles.
- (4) An assessment of cargo binding: the detection of cryo-tomography density features corresponding to cargo.
- (5) The observation of a wide range of cage/coat designs, including novel 5-fold cage vertices.

The main findings of interest in the study are: First, the arrangement of inner-coat protein on spherical vesicles. These are shown to organize as localized patches; this is a nice advance and rationalization of our understanding.

Second, the finding of a diversity of coat/cage structures, including the description of elbow interactions and the 5-fold symmetry centers of the cage vertex. I would like to see a much more thoroughgoing analysis of the varieties of COPII structures. The gold standard study, in this context, is the 2007 J Mol Biol study by Yifan Cheng, Steve Harrison and Tom Walz, in which they used cryo-electron tomography to analyze clathrin-coated vesicles purified from brain. A range of cage designs were defined. The distinctive element in that study is that a series of designs are analyzed individually in terms of their size, geometry/symmetry and architecture. As a result, the paper yields deep insight into clathrin/adaptin architecture.

This level of detail is absent from the present manuscript, and my enthusiasm for publication in NSMB is lowered as a result.

This reviewer rightfully points out that we have not extracted as much information as we could have from our dataset. We focussed on the inner coat arrangement and partly overlooked the outer coat cages. We have now significantly expanded our initial investigation of the outer coat cages which has revealed new insights.

We included additional panels and split Figure 4 into two Figures (4 and 5).

Firstly, as requested, we have carried out a similar analysis of the coat/cages formed by COPII to Cheng et al¹. In Figure 4A we present a gallery of outer cage structures on individual vesicles. We have also modified the text to describe the outer coat features (4-way vertices, 5-way vertices, extra rods, triangular, rhomboidal and pentameric faces) that are typically seen in all cages, and explain how these features give rise to a variety of geometries. Every cage has a unique architecture, resulting in a lack of polyhedral symmetry.

We also expand on this qualitative analysis, going beyond what was done in Cheng et al¹. Based on refined position of cage vertices, we measure individual polyhedral angles. We show that these

continually change within a wide range between 120 and 180 degrees, and strongly correlate with vesicle size (Figure 5A-B).

These analyses strengthen our argument that the architectural heterogeneity of the COPII outer coat allows it to adapt to the continuous range of vesicle sizes that we measured.

In addition, we generated a novel structure of cage vertices that appear to be formed upon convergence of 5 rods. To do this we manually picked them and performed reference free alignments. The average is included in Figure 4D, and shows unambiguous density for 5 Sec13-31 rods that arrange in a 5-fold symmetric manner through interaction of 5 Sec31 N-terminal beta-propeller domains. The EMDB deposition for this map is in progress, with deposition number D_1292137370, EMDB-19879.

It is important to point out that many cages are incomplete. While this can partly be explained by genuine absence of outer coat, many vesicles are thicker than the ice layer, which results in the loss of the outer coat from the top or bottom of the vesicles. Furthermore, the localisation of vesicles adjacent to other features (gold beads, contaminant membranes), could prevent existing outer coat units from being detected. These issues preclude us from being able to reliably quantify completeness.

Minor points:

Cargo:

The analysis of cargo-binding sites draws on prior observations, as the authors acknowledge. The experiments (including Sed5 addition) are nicely done. While it is interesting to see density appear in the cryo-tom maps at known cargo-binding sites on COPII, this is just a demonstration of prior knowledge. These sites were identified originally by high-resolution structural analysis and functional analysis in yeast transport experiments. Consequently, the observation of low-resolution density features at the B-site of Sec24 is not novel.

We agree with the reviewer that we are not describing any novel cargo binding sites. In fact, we chose Sed5 for our experiments because its binding sites were known from X-ray crystallography, facilitating our analysis. The aim of the Sed5 experiment was to assess whether cargo binding is compatible with assembly of the lattice, as this has been a point for controversy in the past.

The absence of density for Sed5 at the A-site is referred to as "interesting", but I really think this should be considered "concerning". This is a bona fide site and interaction validated by budding experiments in yeast. The authors should offer more analysis and information on this contradictory result.

Sec24 'A' site binds to Sed5 peptide YNNSNPF. Previous studies² have shown that this peptide is occluded in the monomeric form of Sed5, and it only becomes exposed when Sed5 is in the Golgi SNARE complex with Bet1, Bos1 and Sec22. As we are adding monomeric Sed5 to GUVs, the poor occupancy of Sec24 A site is expected. This was briefly stated in the manuscript but we have now expanded this section. We have also edited Supplementary Figure 2 to make this more immediately apparent to the reader.

Reviewer #2:

Remarks to the Author:

The work by Pyle and Zanetti is a beautiful follow up to prior work in the Zanetti lab describing the structure of the COPII coat assembled on membranes. Here, they use ER microsomes, as opposed to synthetic GUVs, to demonstrate that yeast COPII largely generates spherical vesicles, as opposed to tubules described in their earlier work. The distribution of the inner COPII coat on vesicles is interesting, suggesting that an intact inner layer is unnecessary for vesicle formation. Studies with the Sed5 cargo are compelling, and the findings shown are logical (although not particularly novel). New insights into the structure of the outer coat are also highlighted. However, the authors should address a few shortcomings prior to publication of this work:

1. The authors claim that vesicle scission mediated by COPII is independent of GTP hydrolysis. However, the incubation period of purified COPII with microsomes and GMP-PNP is extensive (ie., 30 min), much longer than the time required for budding under physiological conditions. It would be best to examine budding following shorter periods of incubation (ie., 1 min or 5 min) to see whether free vesicles are formed. If not, it is likely that the prolonged incubation with active Sar1 results in a non-physiological scission reaction.

We agree that while we are using a non-hydrolysable GTP analogue, we cannot exclude that scission might occur through GTP hydrolysis-independent non-physiological mechanisms and therefore we have toned down our claim that GTP hydrolysis is not necessary for scission.

However, previous results imaging budding reactions of inner coat mutants³ show that such GTP hydrolysis-independent scission does not occur when spherical vesicles are formed on GUVs, incubated for a similar time period as the budding reactions in this study. Instead, these mutants lead to formation of spherical profiles which remain attached through a thin membrane neck, suggesting spontaneous non-physiological scission does not occur on naked GUVs. This demonstrates that GTP hydrolysis-independent scission is induced by some factor which is only present in the microsome prep, and is not a function of incubation time. We have now rephrased the relevant section in the discussion to clarify this.

To comply with the reviewer's request, we have nevertheless prepared grids at different time points. The fastest we could manage without any dedicated time-resolved equipment is a 30 seconds incubation. We additionally froze grids at 2.5 minutes, 5 minutes, and 30 minutes (as the positive control). We collected 10-20 tilt series for each condition and reconstructed tomograms for visual inspection. We saw no COPII-coated membranes at 30 seconds. At 5 minutes, the samples look undistinguishable from the 30 minute control, with several COPII-coated membranes. At 2.5 minutes, there were COPII-coated membranes however less frequent than at 5 and 30 minutes. In all conditions where coated membranes were visible, the vast majority were detached and fully coated. This indicates that scission happens in the first few minutes of our in vitro reactions, and it happens immediately upon coating and deformation, as we did not see a higher proportion of budding vesicles at early time points.

2. The authors should extend their analysis to examine the interaction between the inner and outer layers of the COPII coat. It is unclear why this is not explored in the current manuscript. In particular, are the patches of inner coat observed typically associated with the outer coat? In what manner? Do the inner coat patches form in the absence of the outer coat? Does the heterogeneity of the outer coat correlate with the presence/absence of underlying inner coat?

These are all very interesting questions. The reason why we had not originally discussed this in the manuscript is that we do not detect any fixed relationship between the two coat layers (see below). However, we agree with the reviewer that the relationship between inner and outer coat should be

described, even if it appears to be 'random'. We now add a discussion of this in the manuscript and a new Supplementary Figure 5.

When we plot the positions of inner coat subunits relative to vertices, we do not detect any pattern, indicating that the two layers are not in a 'fixed' arrangement (shown in the 'neighbour plots in new Supplementary Figure 5A).

This could be due to the inner and outer coat lattices being shifted with respect to each other, but maintaining the same orientation (as is seen on coated tubes), or to the two layers being both translationally and rotationally uncorrelated.

To establish this, we have selected and averaged a subset of vertices that all have inner coat present within a restricted neighbourhood region (yellow mask in new Supplementary Figure 5A). If the two layers are rotationally aligned, recognisable inner coat features should appear in the average, below the outer coat vertex. The average map (Supplementary Figure 5B) does not show any recognisable inner coat density, suggesting that the long flexible linker between the inner and outer layers allows for relative shifts as well as rotations. This analysis was also repeated for rods, obtaining the same result (not shown).

As for occupancy, it is important to point out that regions of membranes without inner coat 'patches' might be coated with poorly ordered, or individual units of, inner coat which do not form a lattice (see response to reviewer 3). Likewise, outer coat particles might also not be detected (see response to reviewer 1).

For this reason, we cannot currently assess whether the absence of one layer correlates with the absence of the other with sufficient confidence. Visual inspection of the raw data suggests that the majority of vesicles are mostly coated with both layers (see orthoslices in Figure 1 and Supplementary Figure 1).

3. A mutant form of Sed5 (harboring mutations in its A- and B-sites) should ideally be tested to determine its distribution relative to the inner coat, although this is not absolutely necessary.

We agree that it would be interesting to test these mutants. However, we worry that this experiment would not be very conclusive. Given Sed5 flexibility and the small size of the bound peptide, it is too difficult to classify cargo-bound versus empty Sec24 particles, even with wild type proteins, giving us very little confidence in measurements of cargo distribution.

Reviewer #3:

Remarks to the Author:

Summary: This manuscript examines the structure and organization of the COPII lattice on cell-derived membranes, building upon previous high-resolution tomography results on COPII structure by the same lab on in vitro reconstituted lattices on GUVs. The authors identify a fragmented organization in spherical ER-derived vesicles, which contrasts with the more idealized lattices previously solved on GUVs. The authors use both systems in this manuscript, employing the native-like membranes for insight into COPII organization and then moving to the GUV ordered assembly for higher resolution structures. Biological insights presented in this work include arguments that the inner coat interactions with native lipids and cargo influence curvature, and that the patchy native lattice allows room for cargo binding and presumably a greater diversity of cargoes than the ordered helical assembly on GUVs would.

Overall evaluation: The study is well designed in terms of rationale and methods, making good use of a combination of idealized and more native-like systems to balance biological insight with resolution. In cases where higher resolution is not possible, such as cargo identification, the authors use

reasonable controls (such as the use of subtraction maps against GUVs devoid of cargo). Some key points need to be clarified in the methods, and in general the paper would benefit from greater discussion of previous work and alternative interpretations. I recommend publication following revisions.

Major Points to Address:

1. The microsome system needs to be better described, in particular whether ER-derived microsomes have intrinsic vesicle diameters or curvature patches in the absence of COPII, and whether major protein components present have been identified. Hopefully this was already done in previous publications and just needs to be clarified in the text. If not, it would aid the paper to include mass spectrometry identification of major protein components in the microsomes, for example to confirm that cargo is likely the density described in Figure 3 rather than a cofactor.

We have compared the diameter of COPII coated vesicles to that of uncoated microsomal membranes, and this is already reported in Fig. 1C. This shows that COPII-coated vesicles are significantly more curved than donor membranes, indicating active membrane deformation by the coat. For a fully controlled experiment, we have measured uncoated membrane diameters from negative control budding reactions where COPII was added in the presence of GDP (leading to no coated membranes).

Microsome preparations contain contaminations from many membranes as well as cytosol. Consequently, proteomics analysis by mass spectrometry detected almost 8000 proteins representative of most cell compartments, which is why we had not originally included it in the manuscript.

However, prompted by the reviewer's request, we further analysed the mass spectrometry data specifically focusing on the potential for cargo incorporation, and make the following two observations.

1. We detected many membrane proteins that are residents of the plasma membrane, Golgi apparatus, and endosomes. A subset of any of these protein species present in cells at any time point is likely to be newly synthesised and located in the ER, awaiting for transport. While mass spectrometry cannot discriminate between different membrane compartments, we can assume that a proportion of all these proteins is a potential cargo in our reconstituted system.
2. We specifically looked for proteins that cycle between the ER and the Golgi, supported by data in <https://www.yeastgenome.org/locus/S000001371/interaction>, and Chatterjee et al⁴. We detected a number of proteins that are known Sec24 cargo, including cargo adaptors (Erv25, Erv29, Erv46, Erv41, Emp24, Emp47, Svp26, several Erp proteins), components of the Golgi SNARE complex (Sec22, Sed5, Bet1, Bos1), HDEL receptor (Erd2), and other Sec24 interactors (Yor1, Prm8, Shr3, Rer1, Yip3, Gap1).

It has been shown previously that many of the cargo we identified are concentrated in COPII vesicles with respect to the donor membranes^{5,6}. Therefore, there is sufficient evidence that many potential cargo are present on microsomal membranes to justify the extra density in our maps.

We have now discussed this in the revised manuscript.

2. The new lattice arrangement is a very interesting result, but still leaves the question of whether the native lipids are now resulting in a native-like assembly. I do NOT propose that the authors perform in situ tomography themselves (in my opinion that is unreasonable), but are tomograms available in EMPIAR that they can search and confirm general similarity in COPII patchy coatings of vesicles, or other published evidence they can use to bolster claims of physiological relevance? Perhaps the chlamydomonas dataset from Engel et al, EMPIAR 11756?

We found evidence of COPII in only one tomogram in the dataset suggested by the reviewer. There are other datasets on EMPIAR, but due to lack of annotation and the fact that the original studies do not focus on ER exit sites, we felt it would have required a very long time to reconstruct all tomograms and identify COPII-coated vesicles. However, previous work on the structure of *Chlamydomonas* COPI vesicles⁷ shows COPII in one of their Figure panels (Figure 1D in the original reference). We have prepared the Figure below for the reviewer's benefit to compare our COPII coated vesicles and the COPII coated vesicles in *Chlamydomonas*. In panel A is an overview of an ER-Golgi area in *Chlamydomonas*, where we point to two COPII coated buds. In panels B and B' we include the authors' original figure where they show a complete COPII-coated vesicle in *Chlamydomonas*. In panel C, we show one of the vesicles obtained in this study (scaled to appear at the same magnification as B). While we can't assess the 'patchy' arrangement of COPII in the *in situ* data, the overall appearance of the coat is identical, suggesting that we are looking at structures compatible with physiological arrangements.

We have now added a reference to this in the discussion.

3. The current methods sound like the authors are using denoised tomograms for subtomogram averaging - hopefully there is a missing step in the method description? I would like the authors to confirm that denoised tomograms are used to identify particle positions, but then the particles are extracted from non-denoised tomograms prior to alignment and averaging.

The original manuscript contains a mistake in which we state that IsoNet reconstructed tomograms were used in the initial averaging of the inner coat coordinates in Dynamo. We have rectified this mistake. Furthermore, we confirm we used denoised data only for picking. For subtomogram

averaging, we used RELION v4 and v5, which creates pseudo-subtomograms from raw tilt images and does not use denoised tomograms. We have now clarified this in the methods.

4. This is related to point #3: the authors are deriving meaning from the absence of particles, which requires high confidence in the accuracy of particle identification. IsoNet is a powerful denoiser, but it achieves this goal in part through a de facto averaging along surfaces, which can lead to induced density or lack thereof for membrane-bound particles. As a control, I would like to see in the supplement orthoslices of denoised and non-denoised tomograms for the microsome COPII assemblies, with particle picking comparisons of COPII patches showing that picking is facilitated by denoising but not systematically affected (for example, empty patches of COPII because the denoiser erased them, particularly in areas most heavily affected by the missing wedge artefact). I would also like to see estimates for the accuracy of the particle identification to confirm that expected error would not affect the conclusions - this may be in the rebuttal only if the authors cannot make a robust analysis for publication.

We have added a panel to Supplementary Figure 1 where orthoslices from un-processed tomograms are shown side to side to IsoNet-corrected ones.

Particle picking was initially done manually, and after a template was obtained the particle dataset was expanded by template matching. Particles were detected on the sides but also on the top and bottom of vesicles, when the diameter of these vesicles was smaller than the ice thickness and the vesicles were not proximal to the air-water interface, suggesting that IsoNet and the missing wedge do not make particles disappear in an orientation-dependent manner. Template matching, carried out on non-denoised tomograms, also identified the same particles that were previously manually picked, strengthening our confidence that they were accurate particle picks and not hallucinations by IsoNet.

Visual inspection of tomograms suggests that most of the vesicle membrane is coated with inner coat (as visible from orthoslices), but only a portion of it is picked by template matching, probably because only instances where several inner coat subunits are arranged in a lattice have enough signal to be detected.

We initially extensively tried to align particles picked by randomly oversampling the membrane, similarly to what we did before on tubes, but we could only obtain very low-resolution lattice-like arrangement where sec23 was not distinguishable from sec24, and also the directionality of the lattice was unclear (Figure below: a top view of a STA map obtained by aligning particles extracted from random oversampling of vesicle membranes. Particles were assigned initial angles normal to the membrane, and random in-plane rotations. Reference free alignments detect the presence of a lattice but cannot resolve any features).

This suggested that we need a relatively clean dataset of well-centred particles containing at least 3-4 inner coat subunits arranged regularly in order for them to align.

In summary, we believe inner coat subunits are prominent around membranes, but both targeted picking as well as alignments from randomly oversampled positions fail to identify subunits that are not within relatively extended ordered patches.

Therefore, rather than inferring meaning from the 'absence of particles', we infer meaning from the 'absence of ordered particles'. We have now clarified this in the text by stating clearly that 'empty' patches are most likely coated by disordered inner coat.

With regards to validation: upon picking and reference-free alignments, particles arranged in space in regular lattices. The lattice 'information' was not used for picking, and therefore we can use it as a validation that selected particles are true positives.

List of additional points to consider by line:

- Line 99: Blotting generates shearing forces that are probably strong enough to rip a narrow neck. This doesn't mean this interpretation is incorrect, but please reword to include the possibility of this mechanical perturbation's influence on the result.

We have expanded discussion of this point, as pointed out in our response to reviewer 2.

- Figure 2: please put one of these models on top of an orthoslice, which can show the reader what the blank spaces look like

For consistency within the Figure, we have left Figure 2 unaltered, and added a panel in Supplementary Figure 1D where a back-mapped model is superimposed to an orthoslice.

- Figure 2: is there any data elsewhere, such as mass spectrometry-based stoichiometries, to confirm the finding that the vesicles are only half-occupied by COPII? This would be very helpful to cite in the discussion.

Please see response above. Vesicles are not half-occupied, but rather occupied by partially disordered coat.

- In general the argument regarding the Sed5 and Figure 3 gets rather murky — cargo binding doesn't disrupt the lattice, but the lattice isn't a strong feature in physiologically-relevant lipids based on the ultrastructure? This section could use some attention to tighten the argument.

The argument we tried to make is that small or flexible cargo does not disrupt the lattice, but on native membranes the lattice is disrupted in places, and this is probably attributable to the presence of more bulky proteins.

We make this argument in the discussion and in the final model figure. We have now added a sentence in the results section when we talk about cargo binding to further clarify: *"The lattice disruption we observe on native microsomes is therefore probably due to the presence of more bulky proteins."*

- Section beginning at Line 136: please include more discussion regarding the assumed cargo density (+/- Sed5), for example whether there could be subunit flexibility due to changes in lipids, native cofactors, etc. I agree that the authors' interpretation is the most likely, but other interpretations are possible and should be addressed. Have they been ruled out, and if so why?

The experiment with Sed5 used the same GUV composition as the previous experiments without Sed5. The proteins added were also obtained from identical constructs and purified in the same way. The only difference between the two experiments compared in Figure 3E is the presence of Sed5, which was exchanged into the budding reaction buffer prior to use. While it is not impossible that (undetectable, see Supplementary Figure 2B) contaminants in the Sed5 prep might bind to Sec24 in the Sed5-binding site, we consider that highly improbable and do not think discussion of this possibility would be a valuable addition to the manuscript.

•Line 248 and Figure 4E: have these hinge regions been previously identified, or predicted based on sequence/AlphaFold? Please mention any previous findings or predictions.

The hinges we identify here between beta propeller domains at the boundary between Sec31 and Sec13 (Figure 5D) have not previously been reported. However, previous structural work identified regions of flexibility in the Sec13-31 assembly at various points:

1. At vertices, where angles formed in cuboctahedral versus icosidodecahedral cages obtained by single-particle cryo-EM are different^{8,9}. We detect flexibility at the same point as well, but in a continuous range, rather than vertices assuming a set of defined polyhedral angles. This is noted in the updated Figure 5 A,B (also see response to reviewer 1).
2. At the rod dimerization interface. Here, previously obtained cryo-EM maps (from human proteins) show bent rods (angle of 135°)^{8,9}, while a previously obtained X-ray structure (from yeast proteins) is more flat (165°), as described in Figure 2 of Fath et al¹⁰. All our maps of Sec13-31 rods (from yeast) have angles of 135°, identical to the X-ray structure, and the dimerization interface is well-resolved, indicating it is relatively rigid. We found this observation to be consistent for both rods on tubes³ and on vesicles (this study). We therefore believe that the dimerization interface is not a hinge region, but might display species-specific structural differences.

The hinges we identify here between beta propeller domains at the boundary between Sec31 and Sec13 (Figure 5D) have not previously been reported.

•Lines 253, 262: I think "circles" would be more clear than "lozenges" for the majority of readers

We changed lozenges to rhomboids/rhomboidal where relevant.

•Line 270: "which largely comprise of the ER" - awkward wording. Which are largely comprised, perhaps?

Changed

•Line 332 (Sec 23/24, Sec 13/31, Sar1 purification): the cited paper has a mixture of cleaved and uncleaved His tags. Please clarify whether His tags were present here.

We have clarified that the protocols where the His tag was cleaved were used for both Sec23/24 and Sec13/31. Sar1 always has the His tag cleaved in the cited protocol.

•Section beginning at Line 420 (tomography data collection): please state the number of tomograms collected and the number included in the final particle datasets.

The number of tilt-series was previously stated at the bottom of each paragraph here, we have clarified that each of these tilt series were used to generate a tomogram. We have now described the number of particles used in the final particle datasets in the Subtomogram Averaging section.

- Please state the number of particles used to generate the reference, and confirm that they are not included in the subsequent subtomogram averaging (to maintain independence in halfsets).

We have now stated the number of particles used to generate the Dynamo reference after manual picking. These particles were used in subsequent refinements. Particles which generate an initial model do not have to be discarded from future processing to maintain gold-standard independence, as long as references are sufficiently low-pass filtered. For example, it is standard procedure in single-particle analysis to generate an initial model by running an Initial Model job in RELION and then continue to use these particles for further processing.

- Please add FSC traces to the supplement for all stated resolutions.

We have added FSCs as Supplementary Figure 6.

- Line 499 and elsewhere: please include the resolution of the reference used in template matching, and confirm that final resolution of the subtomogram average surpassed the resolution of the reference (or other approaches to guard against Einstein from Noise artefacts).

We filtered the resolution of the template to 25 Å, i.e. Nyquist frequency of the bin8 tomograms, as recommended by the PyTOM template matching protocol. The resolution of our final map, 14.4 Å, is significantly improved. Furthermore, we found that template matching spontaneously found COPII inner coat subunits arranged in local lattices, minimising the possibility that the picked particles and their poses result from Einstein from Noise artefacts.

- Line 584: I'm assuming that the inner coat must be present for the outer coat to bind, and this is why the positions were not searched independently? Please state this clearly in the methods.

The positions of the outer coat were searched independently from the inner coat for the microsome data. The only assumption we made was that outer coat vertices should be within a certain range of distances from the membrane. We ran template matching on the whole tomograms and accepted peaks that were within masks defining this range.

For Sed5-GUVs, we used the position of the inner coat to set the radial distance and randomly oversample the outer coat layer. The outer coat particles were subsequently aligned and cleaned independently of the inner coat, and in fact the final aligned outer coat has lost any 'fixed' relation to the closest inner coat subunit. We clarified this in the methods.

1. Cheng, Y., Boll, W., Kirchhausen, T., Harrison, S. C. & Walz, T. Cryo-electron tomography of clathrin-coated vesicles: structural implications for coat assembly. *J. Mol. Biol.* **365**, 892–899 (2007).
2. Mossessova, E., Bickford, L. C. & Goldberg, J. SNARE selectivity of the COPII coat. *Cell* **114**, 483–495 (2003).
3. Hutchings, J. *et al.* Structure of the complete, membrane-assembled COPII coat reveals a complex interaction network. *Nature Communications* **12**, 2034 (2021).

4. Chatterjee, S., Choi, A. J. & Frankel, G. A systematic review of Sec24 cargo interactome. *Traffic* **22**, 412–424 (2021).
5. Otte, S. *et al.* Erv41p and Erv46p: New Components of Copii Vesicles Involved in Transport between the ER and Golgi Complex. *Journal of Cell Biology* **152**, 503–518 (2001).
6. Malkus, P., Jiang, F. & Schekman, R. Concentrative sorting of secretory cargo proteins into COPII-coated vesicles. *Journal of Cell Biology* **159**, 915–921 (2002).
7. Bykov, Y. S. *et al.* The structure of the COPI coat determined within the cell. *eLife* <https://elifesciences.org/articles/32493> (2017) doi:10.7554/eLife.32493.
8. Stagg, S. M. *et al.* Structure of the Sec13/31 COPII coat cage. *Nature* **439**, 234–238 (2006).
9. Stagg, S. M. *et al.* Structural basis for cargo regulation of COPII coat assembly. *Cell* **134**, 474–484 (2008).
10. Fath, S., Mancias, J. D., Bi, X. & Goldberg, J. Structure and organization of coat proteins in the COPII cage. *Cell* **129**, 1325–1336 (2007).

Reviewer comments are in black, our responses in green.

Reviewer #3:

Remarks to the Author:

Large overview of the manuscript revisions: in general the rigor of the response to reviewer suggestions is underwhelming and has dampened my enthusiasm for publication in a higher-tier journal like NSMB.

Points still at issue following revision:

Major points 1 and 2 have not been robustly addressed. Uncoated microsome diameters etc. were not measured,

The comparison of diameters of coated vesicles with diameter of donor membranes was reported in the original manuscript (Figure 1C). We pointed this out in our first rebuttal, and regret that this was for the second time misunderstood. The relevant sentence in the original manuscript reads “The microsome-derived COPII coated vesicles are significantly smaller than the donor membranes, as measured in a control sample where GDP was supplemented in place of GMP-PNP, demonstrating that the membrane is being actively deformed by COPII (Figure 1C).” (line 97-100 in current version)

We also paste here the correspondent Figure 1C panel.

and the authors did not propose a different way of showing physiological relevance in the absence of EMPIAR tomography data. The published reference image they included, which was identified as COPII in the eLife paper, appears more ordered than the authors' results.

In the first round of review, this reviewer asked us to show that the coated vesicles we observe when we reconstitute COPII budding from native membranes are physiologically relevant. They suggested to find COPII-coated vesicles in EMPIAR datasets to analyse the structure of the coat in situ. We found two major issues with this suggestion: 1. most EMPIAR datasets are not annotated and we would need to download, reconstruct and analyse all of them. 2. even if we found COPII vesicles in EMPIAR data, we would be unlikely to find enough events to perform subtomogram

analysis of the coat, as none of the published datasets were particularly aiming at visualising ER exit sites, so COPII vesicles will be rare ‘random’ findings.

We instead provided a comparison of tomogram slices from our reconstituted reactions with a published image of COPII in *Chlamydomonas* in situ from Bykov et al.¹, where we think it is clear that the morphology of our in vitro reconstituted coated vesicles is compatible with what is seen in situ. Pasted again below.

Based on comparison of 2D slices from the image above in our first rebuttal, this reviewer now claims that the coat on in situ *Chlamydomonas* vesicles is more ordered than on our in vitro reconstituted ones. We find it very hard to assess ‘order’ from 2D slices and would not have the confidence to support the reviewer’s statement in this regard.

To overcome the lack of annotation, we contacted Wanda Kukulski, author of EMPIAR-11462, which contains tomograms from *S. cerevisiae* cells. She kindly shared her annotations allowing us to identify some tomograms that contain COPII vesicles (a gallery below).

The point of our side-by-side comparison is that, based on morphology and overall appearance, our reconstitutions seem compatible with COPII vesicles in cells. To assess order in *in situ* data, we would need to pick and average particles, which is not achievable with current EMPIAR datasets.

We agree with the reviewer that determining the coat structure in situ would be a very exciting project (which we are indeed pursuing) but it requires a major investment of time and resources and is beyond the scope of the current manuscript.

On the data processing side:

-Some aspects of the processing approach are controversial. Specifically, the use of particles in both the initial reference and the dataset and the use of higher-resolution references to aid particle picking. The authors mention the RELION workflow as requiring the inclusion of particles from the reference in the dataset, perhaps they do not realize there are other options possible within that software environment?

Regardless, these controversial approaches can be validated by the final refinement of a high-resolution structure with features far exceeding the initial reference, and clean FSCs. In this case, the tests do not pass: use of a 25Å reference results in a 14Å map and the FSCs have issues. One of the new FSCs does not cross the x-axis; another has a shoulder that is likely either the mixture of a subset of noise-aligned "particles" with real data or overfitting of flexible protein. These may indicate problems with the data processing, discussed above.

Before tackling all specific points here, we would like to summarise our procedure for averaging inner coat on microsomes, as stated in more detail in the methods, but in addition we include snapshots of the maps used at the various stages. Please note that all alignments in relion are carried out by splitting the dataset into two independent halves by design.

1. we manually picked ~4700 particles from IsoNet-denoised tomograms
2. we assigned initial orientation normal to the membrane (with random in plane rotation angle)
3. we used the manually picked coordinates to extract particles from CTF corrected, non-Isonet treated tomograms.
4. We averaged them to obtain a 'blob'. This had no features due to the random in plane rotations assigned -> Map1
5. The same manually picked dataset was aligned to the 'blob' using dynamo and a low-resolution lattice reminiscent of Sec23/24 spontaneously appeared. -> Map2
6. We further refined the dataset above in relion (using Map2 filtered at 30Å as reference) to obtain Map3

7. We used Map4 (Map3 low-pass filtered to 25Å and masked) as a template for template matching from CTF-corrected, non-isonet treated tomograms, and found ~30000 particles.
8. We run 3D Classification in relion on the template-matched particles using Map3 filtered to 25Å as a reference and selected good particles.
9. We combined the manually picked particles with the selected automatically picked particles.
10. We averaged all of them to obtain Map5.
11. We aligned the full dataset to Map5 (low-pass filtered to 25Å) using relion.
12. Upon rounds of particle selection (also excluding particles with no regularly arranged neighbours) we obtained two half maps that converged to a resolution of 14Å, where we unambiguously fit Sar1/Sec23/Sec24 subunits (final map as in Figure 2).

This reviewer repeatedly criticises the fact that the particles that contributed to the starting reference also contribute to the final map. We believe this is standard practice, as in the published relion tutorial (https://relion.readthedocs.io/en/release-5.0/STA_tutorial/index.html). We are very familiar with relion processing routines, having contributed to its development (Euan Pyle is co-author of two and Giulia Zanetti of one manuscript describing developments for subtomogram averaging in relion ^{2,3}).

We'd like to note, as clearly outlined in our methods and above, that the reference we used for picking was derived from alignment of manually picked particles to a starting reference obtained by assigning random in-plane rotations to the picked particles (i.e. a blob), meaning that no external map was ever used in the processing and that any feature has spontaneously emerged from the data over the course of our processing.

Regarding more specific points:

“use of a 25Å reference results in a 14Å map”

Given our data processing parameters, this is equivalent to an information gain over 12 Fourier pixels, almost 20% of the Fourier domain. As per ‘gold standard’ procedures, this gain originates from comparing two independent half datasets, so we believe there is no problem there. We note the FSC corresponding to the map in question is ‘clean’, according to the reviewer’s assessment.

“One of the new FSCs does not cross the x-axis”

We believe the reviewer refers to the FSC of the Sed5-bound outer coat rod map (Supplementary Fig 6F). Once crossed, the curve remains well-below the 0.143 threshold until Nyquist. There is a slight 'peak' at 1/0.15 Å, but that remains below 0.143 and the curve goes back towards zero, so we do not believe that is of any concern.

“a shoulder that is likely either the mixture of a subset of noise-aligned “particles” with real data or overfitting of flexible protein”

As far as we are aware, shoulders in the FSC (i.e. uneven distribution of amplitudes) can be caused by many reasons, for example non-uniform distribution of defocus values. As the reviewer suggests, a shoulder could possibly also be caused by flexibility outside the aligned 'core' particles, as this would lead to good correlation of higher resolution features in the half maps, but poorer correlation of surrounding material at lower resolution. Although we don't know whether this is the case here, such an occurrence would not be unexpected in our type of sample and we do not think it is the symptom of a problem.

In general, while we don't claim our datasets are perfect, we firmly believe we have processed them following rigorous and well-controlled routines and have obtained maps which agree with expectations given known structures, and which we interpret to levels that are appropriate to their resolution. In all cases, the estimated resolution agree very well with the map 'appearance'.

-IsoNet and particle picking validation was not well addressed in the rebuttal and revision. Confirming that manual picks from controls are also found by IsoNet-facilitated picking is a poor validation because particles easily seen by the naked eye would be expected to come through any processing approach.

We regret the recurrent misunderstanding. As stated in our previous rebuttal and in the manuscript, Template matching for the inner coat on microsomes was performed on non-denoised and non-IsoNet-corrected tomograms.

Independence of orientation and regularity are certainly good, but not enough to validate a neural network that can impose regularity as part of its algorithm. This would not be a problem for a structure solve, but it is a problem for deriving meaning from the positions and orientations of particles. Stronger discussion of strengths and weaknesses, and tempering of conclusions based on them, is warranted.

The reviewer is concerned that the particles are not real and the presence of a regular lattice is an hallucinated feature. Template matching spontaneously found particles at positions and orientations related by regular neighbourly relationship.

The fact that template matching was carried out on non-IsoNet-corrected data means the lattice information cannot have been 'hallucinated' (as no neural network that can impose regularity as part of its algorithm was used in the process beyond the initial manual picking).

We believe deriving meaning from the regular arrangement of particles is therefore valid.

To further support our belief that the lattice is real, we insert below some images where small lattice patches are clearly visible on grazing slices through non-isoNet corrected data (left), as well as IsoNet-corrected (right).

-The authors are still not making the distinction between the number of tomograms collected/reconstructed and number of tomograms contributing to the final analysis.

We now added the full information:

Number of manually picked particles: 4697

Number of PyTom picked particles: 29496

Number of Tomograms PyTom picked particles in: 475

Number of tomograms contributing particles for the final structure: 352

Number of particles in final structure: 12143

Regarding the conclusions drawn from the structures, several points have not been addressed adequately:

-Shearing forces as a result of blotting are still not mentioned in the manuscript as a possible contributor to scission.

We changed:

“While we do not perform any centrifugation or mechanical perturbation aside from gentle pipetting to prepare our samples, we can’t exclude scission is triggered by non-physiological mechanisms.”

to

“While we do not perform any centrifugation or mechanical perturbation aside from gentle pipetting and blotting to prepare our samples, we can’t exclude scission is triggered by non-physiological mechanisms.”

-Hinges that were not previously indicated in other publications should not be newly identified here from low-resolution structural data.

We respectfully disagree. The reviewer refers to hinges in the outer coat rods, which we identified by classifying subsets of rods that have neighbouring vertices at different positions (Figure 5D and supplementary video). We obtain averages from our analysis where we can unambiguously fit the known domains of Sec13/31. Our description and interpretation of their movement are appropriate to the resolution we obtain, as we do not indicate molecular interactions or specific regions that serve as hinge, but simply point to the presence of a hinge between two domains.

-Sed5 identity caveats are still missing from the manuscript

The reviewer is concerned that the extra density we see associated with the Sed5-binding site in the in vitro reconstituted coat in the presence of Sed5 (Figure 3 C-F) might be a non-specific contaminant. In our previous rebuttal, we stated that: *“The experiment with Sed5 used the same GUV composition as the previous experiments without Sed5. The proteins added were also obtained from identical constructs and purified in the same way. The only difference between the two experiments compared in Figure 3E is the presence of Sed5, which was exchanged into the budding reaction buffer prior to use. While it is not impossible that (undetectable, see Supplementary Figure 2B) contaminants in the Sed5 prep might bind to Sec24 in the Sed5-binding site, we consider that highly improbable and do not think discussion of this possibility would be a valuable addition to the manuscript.”*

We remain of the view that the discussion of this is not of any value. However, to comply with the reviewer request, we now add it to the results section:

“We note that, given the purity of the Sed5 prep (Supplementary Figure 3) and the absence of any additional differences with the cargo-less budding reaction, it is highly likely that the extra density we see corresponds to Sed5.”

-The authors acknowledge that the inner coat is not being identified well in the particle picking. In this case, how can the authors be sure that the minority of inner coat particles they are identifying aren't inherently different, thus making them amenable to picking? And how can they have confidence that the inner coat isn't necessary for COP II outer coat when they are having trouble identifying the inner coat in the first place?

We identify a subset of inner coat particles, those that are ordered, indeed because they are more easily detected and aligned, and we moreover filter out any potential ‘lone’ particles with our neighbour analysis, as stated in the methods. This is done on purpose to make sure only real particles are included in the dataset and we acknowledge that there very likely are non-lattice inner coats that we exclude. We include this caveat in our interpretation when we state:

“Visual inspection of tomograms shows that vesicles appear to have an inner coat even where patches are not detected, indicating that ordered patches and un-ordered individual subunits co-exist on spherical membranes (Supplementary Figure 1D).” This does not affect the validity of our interpretation that patches of lattice form. This also means we do not claim that outer coat can exist above naked membrane, there is likely inner coat underneath, as seen in the raw data, but our data does not allow us to conclude whether outer coat can assemble without binding to inner coat or not, so we do not discuss this.

-The statements the authors are making about the inner coat's positioning are too strong considering they are missing the majority of the data.

As above, we acknowledge we are only looking at the ordered subset of inner coats. If we included ‘lone’ particles, we wouldn’t have a reliable way to filter out the ‘junk’.

1. Bykov, Y. S. *et al.* The structure of the COPI coat determined within the cell. *eLife*

<https://elifesciences.org/articles/32493> (2017) doi:10.7554/eLife.32493.

2. Zivanov, J. *et al.* A Bayesian approach to single-particle electron cryo-tomography in

RELION-4.0. *Elife* **11**, e83724 (2022).

3. Burt, A. *et al.* An image processing pipeline for electron cryo-tomography in RELION-

5. 2024.04.26.591129 Preprint at <https://doi.org/10.1101/2024.04.26.591129> (2024).

We would like to thank Reviewer 4 for their insightful comments. Our responses are below in green. We have tracked edits in the manuscript also in green.

Reviewer #4:

Remarks to the Author:

In this manuscript, Pyle and colleagues determine the structure of COPII assemblies generated using protein from *S. cerevisiae* and native membranes. Previous structures of COPII-coated membranes determined by this group showed that COPII forms a pseudo-helical lattice on tubular vesicles. However, these prior studies reconstituted COPII assemblies using Giant Unilamellar Vesicles (GUVs) generated from purified lipids. These tubular vesicles appear distinct from the spherical COPII vesicles that have been seen in cellular tomograms. To address this potential problem, the authors here reconstitute COPII assemblies using native membranes purified from yeast ER membranes (i.e. microsomes), and find that these assemblies take on a spherical morphology that more closely resembles those seen in situ, rather than those assemblies from GUVs. The authors determine the structures of the inner COPII assembly, and the outer coat centered on either vertices or rods; each of these structures were determined for assemblies reconstituted from microsomes or Sed5-enriched GUVs.

As this manuscript has already gone through at least one round of revision, I will mainly focus my review on the technical aspects of the subtomogram averaging (STA) used to determine the structures presented.

With regards to processing workflow and dataset size, (i.e. number of tomograms collected/used and remaining subtomograms) I think at least a supplementary table would make things easier to follow; a processing workflow diagram showing the steps of the workflow and packages used would be better.

We have added a supplementary table as requested. We report several maps which were obtained with different workflows and we think it would be confusing and impractical to add detailed flowcharts for all of them, and therefore refer the reader to the methods.

In a previous round of review, reviewer 3 referred to some aspects of the processing approach here as “controversial”. Specifically, this statement seems to refer to two things: using particles from the initial reference in the full dataset; and using a “higher-resolution” reference for particle picking.

On the first point, I don’t think I’ve ever seen a STA manuscript that uses an independent subset of the data purely for picking. While there may be some issues with the so-called “attraction problem” where a particle correlates against itself in the averaged reference, this likely has a minor effect in the dataset sizes used here; the use of a maximum likelihood algorithm (i.e. RELION) is more likely to bias the results (see: <https://doi.org/10.1016/j.jsb.2018.08.002>). Viewing this from first-principles, this

point is largely moot as every iterative alignment algorithm at the core of STA or single particle analysis (SPA) uses references generated from the particles.

We fully agree with the reviewer.

On the second point, I think “higher-resolution” is a bit poorly defined to be used in a cogent argument. Any useful template for template matching must be “higher-resolution” than the raw tomographic data and isotropically-resolved ; STA is effectively a pre-requisite. A better way to think about this is what resolution limit is reasonable for template matching without imposing too much reference bias. As I noted in the point above, this is again a fundamental problem of iterative alignment. Here, there are no hard-and-fast rules, but best practices include so-called “gold standard” processing using half maps and low-pass filtering references. “Gold standard” is where the dataset is split in half, and each is refined independently using independent references. However, this splitting is typically performed later in the processing pipeline after particle picking is performed (e.g. 3D auto-refine in RELION); the template matching approach used here follows this. As for low-pass filtering, the ideal approach is to use the lowest resolution information possible. This is rarely known a priori, so another approach is to determine if the resolution of the map is sufficiently beyond the low pass filter limit used for alignment. In the case cited here of a 25Å map yielding a 14Å structure, we can estimate the increase in information content in Fourier space. This shows the information is at 0.040 1/Å and 0.071 1/Å respectively, which is an approximate 1.78 fold increase in information in one dimension and a 5.6 fold increase in 3D information. Looking at the FSC plots in supplementary figure 6, the FSCs of all structures is still at ~1 at the low pass filter cutoff at 0.04 1/Å; this seems completely reasonable since high-resolution refinement will typically push this much further.

We agree with the reviewer, who is making very similar arguments to our previous response to reviewer 3.

I do have some concerns with the FSC plots shown in supplementary figure 6 C and F. For C, the FSC seems to only pass 0 for 1 Fourier shell while in F it never passes 0. For a well-determined structure the FSC should reach 0 and oscillate around 0, as true noise shouldn't correlate; correlation past the estimated resolution can be a sign of overfitting. This ideal behavior is seen in all the other FSCs presented, which may suggest some issue specifically in the outer coat rod processing. I don't agree with Reviewer 3's notion that a shoulder here is related to noise alignment, as I don't see any theoretical basis to this argument. What seems likely to me for C and F are that there are duplicate particles between the halfsets. While the authors mention removing duplicates, I can imagine RELION pushing particles towards neighboring “shiny” particles, effectively resulting in duplication. In any case, this issue needs to be resolved; suggesting that the FSC staying below 0.143 is sufficient is not a reasonable argument.

The reviewer is correct in that sometimes duplicate particles can generate spurious correlation in FSCs. However, their observation that this effect is only seen in FSCs of rods made us realise that the effect in our case is due to the rod-shaped mask we had

used for postprocessing both rod datasets. While smooth, this mask probably still contains features that introduce a correlation in Fourier space and inflate the high resolutions.

We recalculated the FSCs with a smooth spherical mask, and our FSCs now look healthy (see updated Supplementary Figure 6). The nominal resolution is slightly worse (11.8 Å instead of 11.2 Å for the rod on microsomes and 9.5 Å instead of 8.7 Å for the rod on Sed5-GUVs) as more background noise is now included in the two half maps, but this does not change our interpretations at all. We now report the new FSCs in Supplementary Figure 6 and update resolution values in the manuscript. Please note that we deposited (and used for figures) rod maps obtained before postprocessing, so the use of a new mask to calculate the FSC does not warrant any modifications to figures (beyond Suppl. fig 6) or updates in EMDB deposition.

For the point on validating manual picking, I don't think there's any good approach for this as real data does not have a "ground truth". Ultimately, 3D classification is the only practical approach for removing false positives, with the caveat that it is also not perfect. I think reviewer 3's issues with neural network hallucinations are a non-issue, as IsoNet data was not used for image processing.

We agree with the reviewer. We would also add that filtering particles according to the position of their neighbours is a very good way to eliminate most false positives, despite also reducing the number of true positives, as the likelihood of neighbours being in a set geometrical relationship 'by chance' is very low.

In regards to the hinges described in Fig 5, I am a bit unclear about some of the data being shown. For Fig 5A, are the density maps simply illustrations of how rods can bend around vertices, or are they actually averaged from the data? As for the hinges described in C and D, this seems like a fair comparison, but more information is needed. Specifically, how many particles in each class, resolution of each class, and whether the overlaid maps shown are filtered to the same resolution.

The semi-transparent maps in Figure 5A and 5C are meant to be illustrations of many possible rod poses, and they were manually placed to aid visual interpretation of the neighbour plot. We have now clarified this in the Figure legend. We have added the requested information regarding the maps in Figure 5D to the methods section.

Overall, I think the processing pipelines used in this manuscript are pretty standard for the field. While I have described several points that need clarification, I don't see these issues substantially affecting the findings in this manuscript.